



# CM2Mc-LPJmL v1.0: Biophysical coupling of a process-based dynamic vegetation model with managed land to a general circulation model

Markus Drüke[1,2], Werner von Bloh[1], Stefan Petri[1], Boris Sakschewski[1], Sibyll Schaphoff[1], Matthias Forkel[3], Willem Huiskamp[1], Georg Feulner[1], and Kirsten Thonicke[1]

[1]Potsdam Institute for Climate Impact Research, Member of the Leibniz Association, P.O. Box 60 12 03, 14412 Potsdam, Germany
[2]Humboldt Universität zu Berlin, Unter den Linden 6, 10099 Berlin, Germany
[3]Technische Universität Dresden, Institute of Photogrammetry and Remote Sensing, Dresden, Germany

**Correspondence:** Markus Drüke (drueke@pik-potsdam.de)

**Abstract.** The terrestrial biosphere is exposed to land-use and climate change, which not only affects vegetation dynamics, but also changes land-atmosphere feedbacks. Specifically, changes in land-cover affect biophysical feedbacks of water and energy, therefore contributing to climate change. In this study, we couple the well established and comprehensively validated Dynamic Global Vegetation Model LPJmL5 to the coupled climate model CM2Mc, which is based on the atmosphere model AM2 and

5   the ocean model MOM5 (CM2Mc-LPJmL5). In CM2Mc, we replace the simple land surface model LaD (where vegetation is static and prescribed) with LPJmL5 and fully couple the water and energy cycles using the Geophysical Fluid Dynamics Laboratory (GFDL) Flexible Modeling System (FMS). Several improvements to LPJmL5 were implemented to allow a fully functional biophysical coupling. These include a sub-daily cycle for calculating energy and water fluxes, a conductance of the soil evaporation and plant interception, a canopy-layer humidity, and the surface energy balance in order to calculate

10   the surface and canopy layer temperature within LPJmL5. Exchanging LaD by LPJmL5, and therefore switching from a static and prescribed vegetation to a dynamic vegetation, allows us to model important biosphere processes, including fire, mortality, permafrost, hydrological cycling, and the impacts of managed land (crop growth and irrigation). Our results show that CM2Mc-LPJmL has similar temperature and precipitation biases as the original CM2Mc model with LaD. Performance of LPJmL5 in the coupled system compared to Earth observation data and to LPJmL offline simulation results is within acceptable

15   error margins. The historic global mean temperature evolution of our model setup is within the range of CMIP5 models. The comparison of model runs with and without land-use change shows a partially warmer and drier climate state across the global land surface. CM2Mc-LPJmL opens new opportunities to investigate important biophysical vegetation-climate feedbacks with a state-of-the-art and process-based dynamic vegetation model.



# 1  Introduction

Human activities, including land-use change and fossil-fuel emissions, change the climate and lead to profound changes in the components of the Earth system and their interactions. For example, increasing managed land for agriculture and other human activities not only reduces natural vegetation cover, but also changes how energy, water and carbon is exchanged between land, atmosphere and ocean. However, a functioning biosphere ensures stable energy, carbon and water cycles and hence atmospheric composition and radiative forcing are maintained. While plants sequester carbon dioxide ($CO_2$), they also contribute to water cycling, albedo and roughness length, influencing the exchange of energy on multiple time scales (Green et al., 2017; Chapin et al., 2008; Heyder et al., 2011). These effects can alter regional and global climate, and in turn lead to changes in land vegetation. To address the implications of climate and land-use change on vegetation dynamics and land-atmospheric feedbacks, Earth System Models (ESMs) with embedded dynamic vegetation components are required.

ESMs increasingly incorporate Dynamic Global Vegetation Models (DGVMs) to advance from quantifying only simple fluxes of carbon, energy and water from land to also capturing climate feedbacks which result from changes in vegetation cover due to plant mortality and regrowth (Quillet et al., 2010; Forrest et al., 2020; Viterbo, 2002; Pokhrel et al., 2016; Fisher et al., 2018; Mueller and Seneviratne, 2014; Hajima et al., 2020; Green et al., 2017). Originally, DGVMs were developed as stand-alone vegetation models to quantify climate-change impacts on terrestrial vegetation (Prentice et al., 2007). However, over the last two decades they have evolved into whole-ecosystem models, capturing a wide range of biosphere processes for natural and managed vegetation, and simulating global carbon, energy and water fluxes with a good modeling skill when compared to observation data (e.g. Schaphoff et al., 2018b). Therefore, embedding these whole-ecosystem DGVMs in ESMs allows for quantifying which ecosystem response or change in land use can cause climate feedbacks and could have wider implications for the Earth system in the Anthropocene.

Several modelling attempts have been made over the past two decades to achieve this goal, often coupling a DGVM to the land surface model of ESMs and not directly to the atmosphere itself. Bonan et al. (2003) showed a first implementation of an early version of the LPJ DGVM (Sitch et al., 2003) into a land-surface scheme and hence a coupling to an atmosphere model. Another attempt of coupling a DGVM to a general circulation model (GCM) has been done by Strengers et al. (2010), which used an older version of LPJmL (Bondeau et al., 2007) in its land-surface scheme. In recent years, many state-of-the-art DGVMs, such as JSBACH (Verheijen et al., 2013) and ORCHIDEE (Krinner et al., 2005) have been coupled to GCMs, while the DGVM JULES (Best et al., 2011) was specifically developed to add vegetation dynamics to the Hadley Center ESM (Harper et al., 2018). These model developments have allowed researchers to investigate effects of biophysical and biogeochemical coupling in the Earth system, turning atmosphere-ocean general circulation models (AOGCMs) into ESMs (Eyring et al., 2016; Anav et al., 2013). Recently, ESMs are evolving to include land-use by explicitly simulating crops (e.g., Nyawira et al., 2016; Levis, 2010) and by including full biogeochemical cycling of marine and terrestrial carbon and nitrogen (Hajima et al., 2020).

With increasing process-detail and the number of processes captured in the biosphere components of ESMs rising, new challenges in correctly representing potential feedback mechanisms might arise. This includes error propagation resulting from changes in climate that could be amplified by, e.g., increased tree mortality, which then changes land-surface characteristics





over time (Quillet et al., 2010). Hence, a bidirectional and stable coupling of a DGVM with a full water, energy and carbon
cycle remains a challenge (Forrest et al., 2020; Pokhrel et al., 2016).

In this study, we introduce the biophysical coupling of water and energy fluxes resulting from vegetation dynamics as simulated
by the adapted whole-ecosystem DGVM LPJmL5 (Schaphoff et al., 2018a; Von Bloh et al., 2018) with the Geophysical Fluid
Dynamics Laboratory (GFDL) coupled model CM2 (Milly and Shmakin, 2002) in a coarse resolution setup called CM2Mc
(Galbraith et al., 2011). The flexible modelling system (FMS, Balaji 2002) is used to couple the terrestrial biosphere, modelled
by LPJmL5, to the other ESM model components. In this new model configuration CM2Mc-LPJmL v1.0, LPJmL5 supplies the
variables necessary for the coupling (canopy temperature, canopy humidity, albedo and roughness length), thereby replacing
the original GFDL land surface model LaD (Milly and Shmakin, 2002) in the CM2Mc setup. To accomplish the interactive
coupling between LPJmL5 and CM2Mc, additional quantities which were not part of the stand-alone LPJmL5, e.g. the tem-
perature and canopy humidity, were introduced. Benefits of coupling LPJmL5 include the use of the process-based fire model
SPITFIRE (Thonicke et al., 2010; Drüke et al., 2019), its advanced land use and land management scheme, the representation
of permafrost and a state-of-the-art water cycling (Schaphoff et al., 2018a). By using FMS as the coupling infrastructure we
remain flexible in terms of other ESM components. The coarse CM2Mc model grid enables us to have a relatively fast and
computationally low-cost Earth system model, which allows conducting many model realisations under different land use and
trace gas settings. While CM2Mc uses the relatively old, but fast atmospheric model AM2 (Anderson et al., 2004) in a coarse
resolution setup and the ocean model MOM5 (Galbraith et al., 2011), it will be possible to employ the latest GFDL model
developments in our coupled system in the future.

We do not repeat a full evaluation of the CM2Mc model, which can be found in Galbraith et al. (2011). Rather, the evaluation of
CM2Mc-LPJmL under transient historical conditions focuses on vegetation, historic climate change and the climate variables
temperature and precipitation, because of their strong feedback on the biophysical coupling. In addition, we forced CM2Mc-
LPJmL with historic land-use change to analyse the contribution of crops and managed grasslands to biophysical land-climate
feedbacks.

## 2  Methods

### 2.1  CM2Mc and the GFDL modelling framework

We couple LPJmL5 to the Climate Model 2 (Anderson et al., 2004, CM2) framework developed at the Geophysical Fluid
Dynamics Laboratory (GFDL) including the Modular Ocean Model 5 (MOM5) in a lower-resolution configuration. This model
configuration, called CM2Mc, uses the same code as CM2.1, with slight parameter changes in order to adjust to the coarser
grid (Galbraith et al., 2011). In its original configuration, CM2Mc includes MOM5 and the global atmosphere and land model
AM2-LaD2 or AM2-LaD (Anderson et al., 2004) with static vegetation. The atmospheric resolution is 3° latitude and 3.75°
longitude, making the computation time 10 times faster than CM2, but at the expense of larger biases in the modeling results.
The model components are connected via GFDL's Flexible Modeling System (FMS, Balaji 2002). For our development, we





use the code version 5.1.0 from the MOM5 project's git repository[1]. The model configuration is based on the accompanying test case named `CM2M_coarse_BLING`.

### 2.1.1 The Flexible Modeling System (FMS)

The Flexible Modeling System (FMS) is the coupler between the different model components of CM2Mc and has been devel-
oped at GFDL (Balaji, 2002).[2] FMS is a software framework for supporting the efficient development, construction, execution and scientific interpretation of atmospheric, oceanic and coupled climate model systems. The infrastructure is prepared to handle the data interpolation between various model grids in a parallel computing infrastructure. It standardizes the interfaces between various model components and handles the fluxes between them. The flexibility of FMS allows for the relatively simple exchange of model components.

### 2.1.2 MOM5

CM2Mc employs GFDL's Modular Ocean Model (MOM) version 5 in a nominally 3x3° lateral grid, with 28 vertical levels (Galbraith et al., 2011). Meridional grid resolution increases to a maximum of 0.6° at the equator to allow the explicit sim-
ulation of some equatorial currents. The model uses re-scaled pressure vertical coordinates (p*), with the uppermost 8 layers having a thickness of 10 dbar, which increases with depth to a maximum layer thickness of 506 dbar (Galbraith et al., 2011).
MOM5 utilises the tri-polar model grid of Murray (1996) to avoid a singularity at the North Pole and the use of partial bottom cells for a more accurate representation of bottom topography. Where the grid fails to resolve important exchanges of water between ocean basins, the cross-land mixing scheme of Griffies et al. (2005) is employed. MOM5 in CM2Mc is coupled to the GFDL thermodynamic–dynamic sea ice model (SIS, Delworth et al. 2006). For a more complete description of the model setup, refer to Galbraith et al. (2011).
Within MOM5, the Biogeochemistry with Light, Nutrients and Gases (BLING) model is run. It was developed at Prince-
ton/GFDL as an intermediate-complexity tool to approximate marine biogeochemical cycling of key elements and their iso-
topes (Galbraith et al., 2010).

### 2.1.3 AM2

The atmospheric module in CM2Mc is GFDL's Atmospheric Model version 2.1 (AM2, Anderson et al. 2004). It uses the finite
volume dynamical core of (Lin, 2004), as implemented in CM2.1 (Delworth et al., 2006) with dynamics calculated on a C and D grid. AM2 as used here employs the M30 grid, with a latitudinal resolution of 3° and a longitudinal resolution of 3.75°, with 24 vertical levels. AM2 has time steps of 1.5 hrs for the tracers, 9 mins for dynamics, and 3 hrs for the radiative time step. The coupled model includes an explicit representation of the diurnal cycle of solar radiation. For a more detailed description of the model and its configuration, see Galbraith et al. (2011) and Delworth et al. (2006).

---

[1]https://mom-ocean.github.io/

[2]https://www.gfdl.noaa.gov/fms/





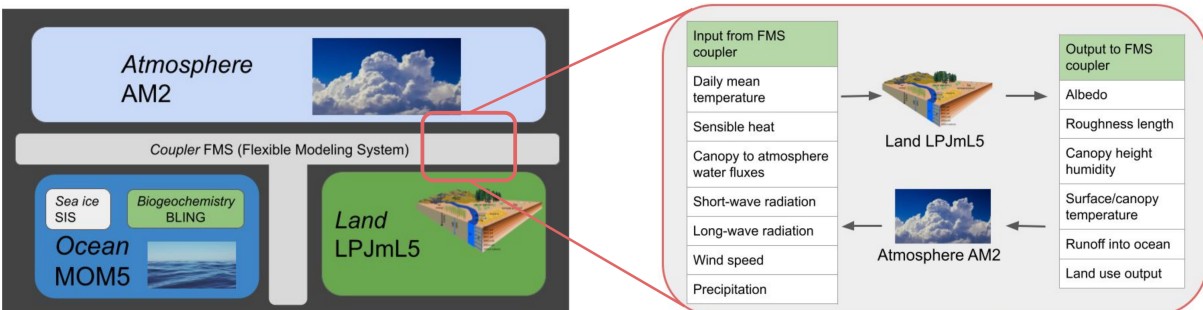

**Figure 1.** Schematic overview of CM2Mc-LPJmL and the variables exchanged between LPJmL5, FMS and AM2.

## 2.2 LPJmL5

The LPJmL5 (Lund-Potsdam-Jena managed Land) DGVM simulates the surface energy balance, water fluxes, carbon fluxes and stocks in natural and managed ecosystems globally and has been intensively evaluated (Von Bloh et al., 2018; Schaphoff et al., 2018a, b). The model is driven by climate, atmospheric $CO_2$ concentration and soil texture data. Since its original implementation by Sitch et al. (2003), LPJmL has been improved by a better representation of the water balance (Gerten et al., 2004), the introduction of agriculture (Bondeau et al., 2007), and new modules for fire (Thonicke et al., 2010), permafrost (Schaphoff et al., 2013) and phenology (Forkel et al., 2014). In this study, we use the updated version of the fire model SPITFIRE as described in Drüke et al. (2019). Since LPJmL5, all LPJmL versions include the nitrogen and nutrient cycle (Von Bloh et al., 2018), which are however deactivated in this study (further adaptions would be necessary to include the nitrogen cycle in the coupled model which is beyond of scope here).

LPJmL5 simulates global vegetation distribution as the fractional coverage (foliage projective cover or FPC) of plant functional types (PFTs) which changes depending on climate constraints and plant performance (establishment, growth, mortality). Plants establish according to their bioclimatic limits (adaptation to local climate) and survive depending on their productivity and growth, their sensitivity to heat damage, light and water limitation as well as fire-related mortality. The interaction of these processes describes the simulated vegetation dynamics in natural vegetation. The model also simulates land use, i.e. the sawing, growth and harvest of 14 crop functional types and managed grassland (Rolinski et al., 2018). The proportion of potential natural vegetation and land-use within one grid cell is determined by the prescribed land-use input.

In standard settings the model operates on a global grid with a spatial resolution of $0.5° \times 0.5°$. However, the actual resolution can be changed according to the spatial resolution of the model input.

To bring vegetation and soil carbon pools into equilibrium with climate, the model is run for a uncoupled spin up time of 5000 years, where the first 30 years of the given climate data set are repeated.



## 2.3 Adapting LPJmL5 to implement into the FMS coupling framework

The coupling software FMS, and hence the atmosphere model, expects a certain set of variables for full dynamic coupling. We consider canopy humidity, soil and canopy temperature, roughness length and albedo as essential variables to allow dynamic vegetation to fully interact with the atmosphere, and describe their implementation in this section. All these variables are

exchanged with the atmosphere on the so-called "fast time step", for which we currently set one hour. Because the offline-version of LPJmL5 simulates carbon and water fluxes only at a daily time step, we introduced a sub-daily time step of the same duration as the fast time step and ensured a diurnal cycle for temperature and humidity which is important to stabilise the atmosphere and the coupled model system (Randall et al., 1991; Kim et al., 2019). These processes included calculations of the water and energy cycles, i.e. surface temperature, evapotranspiration and water stress. Albedo and roughness lengths are

expected to be less dynamic and are thus not dependent on the diurnal cycle. Hence, they are calculated in the original daily time step within LPJmL5, but still exchanged every hour. For ecosystems that are temporarily covered by snow, sublimation is implemented building on the simple snow model in LPJmL5, which also operates at the fast time step. In the fast time step, the coupling variables are sent from LPJmL5 to the FMS coupler. The coupler then provides the synoptic climate variables (temperature, precipitation, radiation) as the input for LPJmL5 in the next (fast) time step.

### 2.3.1 Interface between FMS and LPJmL5

The C main function of LPJmL5 used in the offline version is replaced by a coupler function providing the interface between the internal C functions of LPJmL5 and the Fortran functions of the CM2Mc model. The coupler function is called by FMS on an hourly time step and calls itself the specific update functions of LPJmL5 at the end of each hour, day, month or year, respectively. Ingoing and outgoing data are transferred as array arguments of this function. The mapping of the coarse resolution

of the CM2Mc model to the $0.5° \times 0.5°$ resolution of LPJmL5 is done by the FMS coupler. We found that the FMS land model component must be run at LPJmL5 resolution, which is $0.5°$, so that all model components and the FMS coupler agree on which cells belong to land which to the ocean. This yields slight changes of the land-sea-mask from the original `CM2M_coarse_BLING` setup.

CM2Mc as well as LPJmL5 can use the Message Passing Interface (MPI) to run the simulation in parallel on a compute cluster.

CM2Mc uses FMS to set up a 2-dimensional domain decomposition, i.e. it splits the global grid into rectangular domains which are mapped to concurrent MPI tasks. In contrast, the LPJmL5 grid is represented by an unsorted 1-dimensional array of land cells, which is evenly distributed onto the MPI tasks. Since this LPJmL5 grid is not compatible with the FMS grid exchange framework, a small wrapper library for the data exchange between LPJmL5 and FMS domains was developed. The wrapper library is called for the ingoing and outgoing data and the time overhead for this data exchange is negligible. The coupler

function as well as the wrapper library are part of the LPJmL5 distribution.



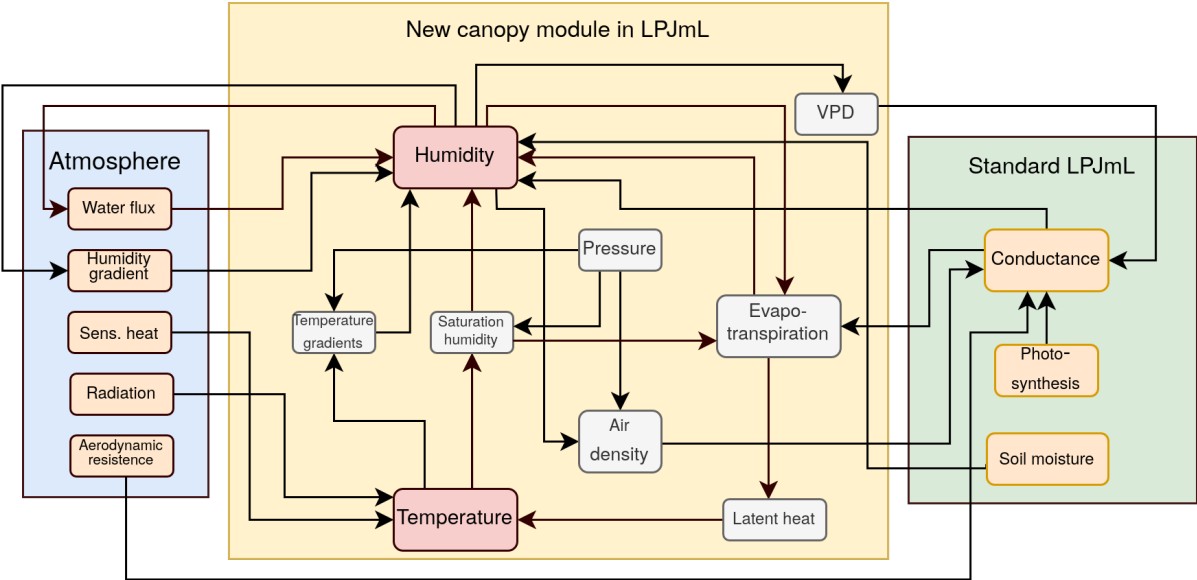

**Figure 2.** Schematic overview of the new Canopy module.

### 2.3.2 New canopy module

The stand-alone version of LPJmL5 does not calculate the essential coupling variables canopy temperature and humidity, which is remedied in the coupled configuration via the addition of a new canopy module. In this new module, the canopy humidity and canopy temperature and some further quantities linked to those variables are calculated (Fig. 2). In this setting, the canopy
layer corresponds to the lower boundary for the temperature in the atmosphere. The atmospheric diurnal cycle as well as the seasonal changes depend on the surface energy balance. The canopy humidity, on the other hand, is the lower boundary for the atmospheric humidity and hence, sets the moisture content and the amount of precipitation in the atmosphere, as well as the potential for evapotranspiration on the surface.

In the stand-alone version of LPJmL5 climatic input is prescribed, and therefore calculations of processes and fluxes, such
as evapotranspiration, do not feed back to the atmosphere. In the coupled version, however, a small perturbation in a positive feedback loop can influence the climate and push the process towards an even larger perturbation. Therefore, special attention has to be given to ensure the stability of the model by either ignoring the feedback and implementing a simple, empirical and stabilizing relationship or by increasing the complexity of the implementation, in order to get a more realistic representation of the vegetation embedded in the earth system. The latter was done in CM2Mc-LPJmL by replacing the former simple Priestley-
Taylor approach for calculating potential evapotranspiration $ET_0$ with the more complex and process-based Penman-Monteith evapotranspiration (Monteith, 1965). The Penman-Monteith approximation also accounts for additional parameters, such as





humidity, that were previously not available in stand-alone LPJmL5 (Fig. 2):

$$\lambda ET_0 = \frac{\Delta(R_n - G) \cdot + 86400 \cdot \frac{\rho_a C_p(e_s^0 - e_a)}{\tau_{a\nu}}}{\Delta + \gamma(1 + \frac{\tau_s}{\tau_{a\nu}})},$$
(1)

where $\lambda$ is the latent heat of vaporization of 2.45 MJ kg$^{-1}$, $ET_0$ is the evapotranspiration in mm s$^{-1}$, $\Delta$ the slope of the

vapor pressure curve in kPa °C$^{-1}$, $R_n$ the net radiation at the surface in MJ m$^{-2}$, $G$ the soil heat-flux density in MJ m$^{-2}$, 86400 the conversion factor from seconds to daily values, $\rho_a$ the air density in kg m$^{-3}$, $C_p$ the specific heat of dry air ($1.013 \cdot 10^{-3}$MJ kg$^{-1}$°C$^{-1}$), $e_s^0$ the saturated water vapor pressure in kPa, $e_a$ the actual water vapor pressure in kPa, $\tau_{a\nu}$ the bulk surface aerodynamic resistance for water vapor in s m$^{-1}$ and $\tau_s$ the canopy surface resistance in s m$^{-1}$. $\gamma$ is the psychrometric constant and is calculated as:

$$\gamma = \frac{C_p P}{\mu \lambda} = 0.000665P,$$
(2)

where $P$ is the atmospheric pressure at the surface in kPa, and $\mu$ the ratio of molecular weight of water vapor to dry air, which is 0.622. Eq. 1 uses the non-waterstressed canopy conductance $g_p$ in mm s$^{-1}$, which is the reciprocal of the canopy surface resistance. $g_p$ was also slightly changed, compared to Schaphoff et al. (2018a) in order to include climate feedbacks. Following Medlyn et al. (2011), we included a PFT-specific stomatal conductance parameter $g_1$ (as defined in De Kauwe et al., 2015) and

the vapor pressure deficit ($D$).

$$g_p = \frac{1}{r_s} = g_0 + 1.6(1 + \frac{g_1}{\sqrt{D}})\frac{A_{dt}}{p_a},$$
(3)

where $g_0$ (mm s$^{-1}$) is a PFT-specific minimum canopy conductance scaled by FPC, occurring due to other processes than photosynthesis. $p_a$ is the ambient partial pressure of $CO_2$ in Pa and $A_{dt}$ denotes the daily net daytime photosynthesis. $D$ (in Pa) can be obtained by the canopy humidity $q_{ca}$ and the saturation humidity $q_{sat}$:

$$D = q_{sat} - q_{ca}.$$
(4)

The newly calculated potential evapotranspiration, using $g_p$, is then also used in several LPJmL5 routines (e.g. bare soil evaporation or interception) instead of the equilibrium evapotranspiration ($E_q$), which was based on the Priestley-Taylor formula (Schaphoff et al., 2018a).

As a next step, we calculate the water-stressed transpiration $E_{tr}$, by using the supply-demand functions of LPJmL5 as follows:

The demand is calculated by the newly implemented potential evapotranspiration (Eq. 1, corrected by the fraction used for interception) and the supply is driven by vertical root distribution and phenology (as in Schaphoff et al., 2018a). The initial transpiration is then a function of the minimum of supply and demand for water. The transpiration is then subtracted from the various soil layers, depending on water availability. If the available water is not sufficient, transpiration decreases. The adjusted transpiration is consequently used in an inverse version of the Penman-Monteith formula in order to calculate the actual canopy

conductance, linked to transpiration $g_{tr}$. The canopy conductance is the reciprocal of the canopy resistance ($g_{tr} = \frac{1}{\tau_s}$).

The total canopy conductance is additionally influenced by the conductance of soil evaporation ($g_e$) and plant interception



($g_i$). Therefore, we use a simple approach taking into account the maximum rainfall interception conductance ($\text{GI}_{\text{MAX}} = 10$ mm s$^{-1}$) and by considering the fraction of rainfall $i$ stored in the canopy of a biome-dependent rainfall regime (Gerten et al., 2004):

$$g_i = \text{GI}_{\text{MAX}} \cdot i \cdot Pr/E_q \cdot f_v \tag{5}$$


where $f_v$ is the vegetated grid cell fraction. The soil-evaporation conductance is calculated for the non-vegetated area of a grid cell and depends on the maximum soil conductance ($\text{GE}_{\text{MAX}} = 10$ mm s$^{-1}$, Huntingford and Monteith 1998), and an empirical scaling factor for the dependency of soil-evaporation conductance on soil-water status ($\alpha_0 = 10$, Zhou et al. 2006):

$$g_e = (1 - f_v) \cdot \text{GE}_{\text{MAX}} \cdot \exp(\alpha_0 \cdot (w_{evap} - 1)) \tag{6}$$


where $w_{evap}$ is the soil water content relative to the water holding capacity available for evaporation defined for a certain soil depth (Schaphoff et al., 2018a).

We then calculate the total canopy conductance $g_c$ by adding $g_{tr}$, $g_i$, $g_e$ and $\tau_{a\nu}$ following Milly and Shmakin (2002).

$$g_c = \frac{\rho_a}{\frac{1}{(g_{tr} + g_i + g_e)} + (1 - \beta) \cdot \tau_{a\nu}}, \tag{7}$$

where $\beta$ is the water available for photosynthesis:

$$\beta = \min\left[\frac{W_r}{0.75 \cdot W_r^*}, 1\right], \tag{8}$$


with $W_r$ as the actual soil water and $W_r^*$ as the maximum available soil water. The increment of the canopy humidity $q_{\text{ca}}$ is then calculated as following, using $g_c$:

$$\Delta q_{\text{ca}} = \frac{ET - q_{\text{flux}} + \Delta_q \cdot g_c \cdot \Delta_t}{\frac{de}{dq} + \rho_a \cdot g_c}, \tag{9}$$

where $q_{\text{flux}}$ is the water flux from the canopy layer to the atmosphere, provided by the FMS coupler, $\Delta_q$ the gradient of the saturation pressure over the temperature, $\Delta_t$ the difference of the actual surface temperature and the temperature of the previous time step and ET the final evapotranspiration, consisting of transpiration, evaporation, interception and sublimation from surface or vegatation into the canopy layer. $\frac{de}{dq}$ is the evaporation–humidity gradient.


It was further necessary to implement the calculation of surface/canopy temperature within LPJmL5, therefore requesting major adaptions to the energy cycle in LPJmL5. Stand-alone LPJmL5 calculates the temperature of different soil layers by employing a temperature transport scheme and taking into account air temperature as climatic input. In CM2Mc, however, the energy balance is calculated on the surface and then passed to the coupler and the atmosphere. Therefore, we had to implement this energy balance analogously in the coupled version of LPJmL5. While this surface temperature depends on several inputs from the coupler, as for instance radiation, it also uses several variables connected to the water cycle in LPJmL5 (evaporation, sublimation and melted water). In order to calculate the canopy temperature within LPJmL5, we employed a simple energy-



balance formulation and use this temperature as the upper boundary for the temperature of the six soil layers in LPJmL5. The





calculation of heat transfer in the soil layers remains as in stand-alone LPJmL5 via a heat-convection scheme (Schaphoff et al., 2018a). The energy-balance formula for the incremental change of temperature for each time step $\Delta T$ is given as (adapted from Milly and Shmakin, 2002):

$$\Delta T = \frac{K + L - m \cdot LE_f + ET \cdot LE_v - Q_{sn} - H}{C_s \cdot \Delta_t}, \tag{10}$$

where $K$ is the incoming net short-wave radiation in $\mathrm{W\,m^{-2}}$, $L$ the outgoing net long-wave radiation in $\mathrm{W\,m^{-2}}$, $m$ the melted ice transformed to water in $\mathrm{kg\,m^{-2}s^{-1}}$, $LE_f$ the latent heat of the conversion of ice into water in $\mathrm{J\,kg^{-1}}$, $LE_v$ the latent heat of the conversion of water into vapor in $\mathrm{J\,kg^{-1}}$, $Q_{sn}$ the released energy by snow in $\mathrm{W\,m^{-2}}$, $H$ the sensible heat provided by FMS in $\mathrm{W\,m^{-2}}$, $C_s$ the heat capacity of the soil in $\mathrm{J\,kg^{-1}}$ and $\Delta_t$ the fast time step duration in seconds. In this implementation, the boundary temperature to the soil layers and the canopy temperature are the same as in LaD (Anderson et al., 2004).

The heat balance of snow is calculated as for the soil layers (see Schaphoff et al., 2018a) where snow temperature changes ($\Delta \mathrm{T}_{\mathrm{snow}}$) depend on the thermal conductivity ($\lambda_{\mathrm{snow}} = 0.2\ \mathrm{W\,m^{-2}\,K^{-1}}$) and heat capacity ($c_{\mathrm{snow}} = 630000\ \mathrm{J\,m^{-3}\,K^{-1}}$) of snow as follows:

$$\frac{\Delta \mathrm{T}_{\mathrm{snow}}}{\Delta t} = \frac{\lambda_{\mathrm{snow}}}{c_{\mathrm{snow}}} \cdot \frac{\mathrm{T}_{\mathrm{air}} + \mathrm{T}_{\mathrm{soil}_{[0]}} - 2 \cdot \mathrm{T}_{\mathrm{snow}}}{\Delta z_{\mathrm{snow}}^2}, \tag{11}$$

and heat flux from snow ($Q_{\mathrm{snow}}$) is calculated:

$$Q_{\mathrm{snow}} = \lambda_{\mathrm{snow}} \cdot \frac{(\mathrm{T}_{\mathrm{snow}} - \Delta \mathrm{T}_{\mathrm{snow}})}{z_{\mathrm{snow}}}, \tag{12}$$

where $z_{\mathrm{snow}}$ is the snow depth, $\mathrm{T}_{\mathrm{air}}$ is the air temperature and $\mathrm{T}_{\mathrm{soil}_{[0]}}$ is the soil temperature of the first layer.

### 2.3.3 Albedo and roughness length

Albedo ($\beta$), the average reflectivity of the grid cell, is calculated as in Schaphoff et al. (2018a), based on a first implementation by Strengers et al. (2010) and later improved by considering several drivers of phenology by Forkel et al. (2014):

$$\beta = \sum_{\mathrm{PFT}=1}^{n_{\mathrm{PFT}}} \beta_{\mathrm{PFT}} \cdot \mathrm{FPC}_{\mathrm{PFT}} + F_{\mathrm{bare}} \cdot (F_{\mathrm{snow}} \cdot \beta_{\mathrm{snow}} + (1 - F_{\mathrm{snow}}) \cdot \beta_{\mathrm{soil}}) \tag{13}$$

where the albedo for bare soil $\beta_{\mathrm{soil}}$ is defined as 0.3 and for snow $\beta_{\mathrm{snow}}$ as 0.7. $\beta_{\mathrm{PFT}}$ is calculated for each PFT depending on the foliage projective cover (FPC) and the stem, litter and leaf albedo of the respective PFT. The value for each parameter is as in Schaphoff et al. (2018a). $F_{\mathrm{snow}}$ and $F_{\mathrm{bare}}$ are the snow coverage and the fraction of bare soil, respectively. Water bodies as lakes and rivers have a constant albedo value of 0.1.

Roughness length $z_{0m}$ is calculated according to Strengers et al. (2010):

$$z_{0m} = z_b \exp\left(-\sqrt{\frac{1}{d}}\right) \tag{14}$$



and

$$d = \sum_{i=1}^{n_{\text{PFT}}} \frac{\text{FPC}_i}{\left[ \ln \left( \frac{z_b}{z_{0m}^i} \right) \right]^2}, \tag{15}$$

where $z_b$ is the height of the boundary layer in stable conditions, set to 100m (Ronda et al., 2003), $z_{0m}^i$ is the PFT-specific
roughness length, and $\text{FPC}_i$ the foliage projective cover of each PFT, respectively.

### 2.3.4   Further changes in the coupled LPJmL5

For a global model we also need to consider Antarctica, which has not been part of the standard grid of the stand-alone LPJmL5
modelling configuration. It was implemented in a simplified approach, and will be replaced with the Parallel Ice Sheet Model
(PISM, Winkelmann et al. 2011) in the future. For now Antarctica is assigned the soil type ice and a constant albedo of 0.7.
The temperature balance is calculated as on the other continents.

In stand-alone LPJmL5, sublimation is subsumed by a constant global value of 0.1 mm per day, likely underestimating the
sublimation at high latitudes. Especially in winter times, we do not expect much evapotranspiration, and hence the sublimation
changes with meteorological conditions and becomes an important process. For this reason, we implemented the calculation of
sublimation $E_s$ by using the formula from Gelfan et al. (2004):

$$E_s = (0.18 + 0.098u)(e_s - e_a), \tag{16}$$

where $u$ is the wind speed in $\text{m s}^{-1}$ from the coupler, $e_s$ the saturated vapor pressure in mbar and $e_a$ the air vapor pressure in
mbar.

Furthermore, first test runs of the coupled models proved the need to tune some LPJmL5 PFT-specific parameters: We increased
the effective rooting depths of the tropical-tree PFTs to 2.3 m in order to counter a negative AM2 precipitation bias in northern
South America. Therefore, we increased the beta-value of each tropical tree PFT describing their vertical fine root distribution
in the soil column from 0.96 as in Schaphoff et al. (2018a) to 0.99 in this study.

### 2.4   Model setup and forcing

In the stand-alone version, as well as in the coupled version, LPJmL5 is forced with gridded soil texture data (Nachtergaele
et al., 2009). Global atmospheric $CO_2$ values are from Mauna Loa station data (Le Quéré et al., 2015) and land-use information
are from Fader et al. (2010). The fire module SPITFIRE (Thonicke et al., 2010) requires human population density as input,
which is taken from Goldewijk et al. (2011), as well as lightning flashes which are taken from the OTD/LIS satellite product
(Christian et al., 2003). In the coupled LPJmL5 version, we activated permafrost, the new phenology and SPITFIRE using the
vapor pressure deficit as the fire danger index (Drüke et al., 2019). The nitrogen-cycle, which is part of LPJmL5 (Von Bloh
et al., 2018), was deactivated in this study. Running in the coupled model, LPJmL5 receives climatic input as for instance
temperature, precipitation and radiation from the coupler interactively.

For the stand-alone LPJmL5 spin-up we used the climate data (temperature and precipitation) from the Land Data Assimilation





System (GLDAS, Rodell et al. (2004)). The original data has a spatial resolution of $0.25° \times 0.25°$ and a time step of 3 h. We re-gridded the data set to the LPJmL5 resolution of $0.5° \times 0.5°$ and aggregated it to a daily time step. For the spin-up we recycled data from the years 1948-1978 (earliest years available in GLDAS). Short-wave and long-wave radiation was used

from the coupled model CM2Mc, where the vegetation has been calculated by LaD (Milly and Shmakin, 2002).

For the fully-coupled model run we used 20 CPUs for the land and atmosphere calculations and 8 CPUs for the ocean, totalling in 28 CPUs. With these settings, one model year needs roughly 30 min on the PIK HPC cluster (Xeon E5-2667v3 8C 3.2 GHz, Infiniband FDR14). The number of MPI tasks is limited by the coarse resolution of the atmosphere grid. Parts of the atmosphere code can employ hybrid MPI+OpenMP parallelism, but computational costs for LPJmL5 remain unaffected.

## 2.5 Modelling protocol

Soil carbon and vegetation biomass need timescales of hundreds to several thousand years to reach an equilibrium with climate, which would require extremely long spin-up simulations in the coupled model. Hence we produce a first spin-up for 5000 years with the more computational efficient stand alone LPJmL5, using climate input from GLDAS and an earlier CM2Mc-LaD run. To bring vegetation, soil and climate into a consistent equilibrium (stand-alone LPJmL5 spin-up and the restart files from

CM2Mc using LaD), we perform afterwards a fully coupled run of 500 simulation years under pre-industrial conditions with land use deactivated. The climate of this run is then used as forcing for another stand-alone LPJmL5 spin-up run of 5000 years, producing restart conditions much closer to the state of the coupled model. This multi-step spin-up approach minimizes the time for the computationally expensive coupled model to reach a stable state.

To account for changed dynamics in the coupled system, the LPJmL5 spin-up is then followed by a coupled spin-up, which

runs for 500 years at pre-industrial and potential natural vegetation (PNV, i.e. without land use) conditions in a fully coupled setting. This fully coupled spin-up is the starting point of the production runs (see Tab. 1), except the pi-CM2Mc-LaD and LPJmL-offline experiments.

As a baseline run, we complete another 250 simulation years under pre-industrial PNV conditions in addition to the 500 simulation years of the coupled spin-up, totalling in 750 simulation years with the same settings (pi-Control experiment).

The transient run (TR) with variable land-use and forcings is performed for the years 1700 until 2018, using historic land-use data from 1700 onward; the concentration of greenhouse gases, solar radiation, ozone concentrations and amount of aerosols in the atmosphere are kept constant at pre-industrial conditions until 1860 and then vary according to historic data. From 2004 onward, solar radiation, ozone and aerosols are kept constant due to missing data.

Similar to the TR experiment, we conduct two more experiments in order to investigate the impact of climate and land-use

change in CM2Mc-LPJmL separately. Both runs are performed for the years 1700-2018, one with transient, historic climate but PNV conditions without land use (PNV experiment) and the other one with transient land-use but pre-industrial climate (LU-only experiment).

Two additional simulation experiments are conducted that did not use the 500 years coupled spin-up: To compare the performance of CM2Mc-LPJmL against the original CM2Mc model under pre-industrial conditions, we conduct a 200-year run of the

CM2Mc model, using the original land model LaD (pi-CM2Mc-LaD) and compare it against pi-Control. Here, we use restart





**Table 1.** Overview over the simulation experiments conducted in this study. All runs, except for pi-CM2Mc-LaD and LPJmL-offline, are performed with CM2Mc-LPJmL. Other forcings include aerosols, non-$CO_2$ greenhouse gases, ozone and the solar constant. In the case of non-transient simulations these are kept constant at their values from the year 1860. Land use can either be transient, i.e. capturing historic changes, or be deactivated.

| Experiment | $CO_2$ [ppm] | Land use | Other forcings |
|---|---|---|---|
| pi-Control | 284 | no | constant |
| TR | 284–408 | transient | transient |
| PNV | 284–408 | no | transient |
| LU-only | 284 | transient | constant |
| pi-CM2Mc-LaD | 284 | no | constant |
| LPJmL-offline | 284–408 | transient | transient |

files provided with the CM2Mc modeling suite. We also perform a transient stand-alone LPJmL5 (LPJmL-offline) run with a deactivated nitrogen cycle (Schaphoff et al., 2018a; Von Bloh et al., 2018) in order to compare the results to CM2Mc-LPJmL.

## 2.6 Model evaluation

Model performance is evaluated in terms of stability and historic climate changes, and the results are compared to pi-CM2Mc-
LaD runs, LPJmL5 stand-alone and observational data. Specifically, our simulation experiments (see Tab. 1) are evaluated as follows: To analyze the stability of CM2Mc-LPJmL, we evaluate temperature and precipitation of the 500-year coupled spin-up run combined with the 250 year pi-Control run (750 years in total).
Climate biases in precipitation and temperature are evaluated by comparing the TR experiment from the years 1994–2003 with global evaluation data sets from ERA5 (Dee et al., 2011). During the years 1994-2003 all forcing in CM2Mc-LPJmL are
transient. Simulated biomass is evaluated by comparing above-ground biomass from the TR experiment with the GlobBiomass gridded data set by Santoro (2018); Santoro et al. (2020). GlobBiomass provides vegetation carbon for roughly the year 2010, hence we compare it to average model data from 2006–2015. The PFT distribution, a measure of vegetation cover, is evaluated by using data from Li et al. (2018) and Forkel et al. (2019), comparing these with results from the TR experiment for the years 2006-2015.
The historical temperature increase is quantified by comparing the transient temperature increase between 1860–2018 of the TR experiment with GISTEMP data (Lenssen et al., 2019). GISTEMP combines various measurements from meteorological stations. To evaluate the impact of changes in atmospheric forcing on the spatial distribution of climate parameters and vegetation, results from the last 10 years of the pi-Control experiment are compared with results from 2006–2015 of the PNV experiment (Section S2). For analysing land-use sensitivity (without variability in the atmospheric forcing), we compare the
last 10 years of the pi-Control and the years 2006–2015 of the LU-only experiment against each other.
In the supplement we further provide a comparison of the results of CM2Mc-LPJmL with CM2Mc-LaD, using an average



over the last 10 years of the pi-Control and the pi-CM2Mc-LaD experiments (Section S3), as well as a comparison with model inter-comparison CMIP5 data (Taylor et al., 2012) and LPJmL5-offline (Section S4).

As evaluation metrics we used the normalized mean error (NME Kelley et al., 2013):

$$\text{NME} = \frac{\sum_{i=1}^{N} |y_i - x_i|}{\sum_{i=1}^{N} |y_i - \overline{x}|}, \tag{17}$$

where $y_i$ is the simulated and $x_i$ the observed value in grid cell i. $\overline{x}$ is the mean observed value. The NME is 1 if the model is as good as using the data mean as a predictor, larger than 1 for worse performance and zero for perfect agreement.

## 3 Results

The evaluation of the model performance is provided in Section 3.1, while the impact of land-use change on the results of the coupled CM2Mc-LPJmL model is analyzed in 3.2.

### 3.1 Model performance

Here, we evaluate the performance of CM2Mc-LPJmL against climate and biosphere observations, by first looking into the long-term stability of global mean surface temperature (referred to as temperature, hereafter) and precipitation (Section 3.1.1) from the pi-Control experiment, before evaluating the historic temperature increase of the coupled model, using the TR experiment results. Finally, a detailed analysis of climate (3.1.2 and 3.1.3) and vegetation cover (3.1.4) is provided, also based on the TR experiment.

#### 3.1.1 Model stability

The analysis of the model stability was based on the pi-Control experiment, which ran over 750 years in total (see Section 2.5 for details). Here, we evaluate temperature and precipitation in terms of absolute values as well as rate of change over time and the variability.

After the initial 300 years, the global temperature remains relatively stable at ca. 14.7°C over the remaining simulation period of 400 years with a slight drift of less than of 0.05°C per 100 years (Fig. 3a). The interannual variability in this period is ca. 0.1–0.2°C. The decreasing temperature over most of the 750-year simulation period can be explained by the energy uptake of the ocean, since deep ocean layers are not yet in equilibrium. The average precipitation follows a similar trend as temperature and reaches a relatively stable state at around 2.88 mm/day after ca. 400 years, changing less than 0.01 mm/day over the remaining period (Fig. 3b). The interannual variability is 0.01–0.02 mm/day.

#### 3.1.2 Temperature evolution over the historical period

The temperature evolution over the historical period, hence the climate sensitivity to changes in atmospheric forcing, is evaluated by comparing the transient temperature increase in the period 1880–2018 of the TR experiment to GISTEMP evaluation



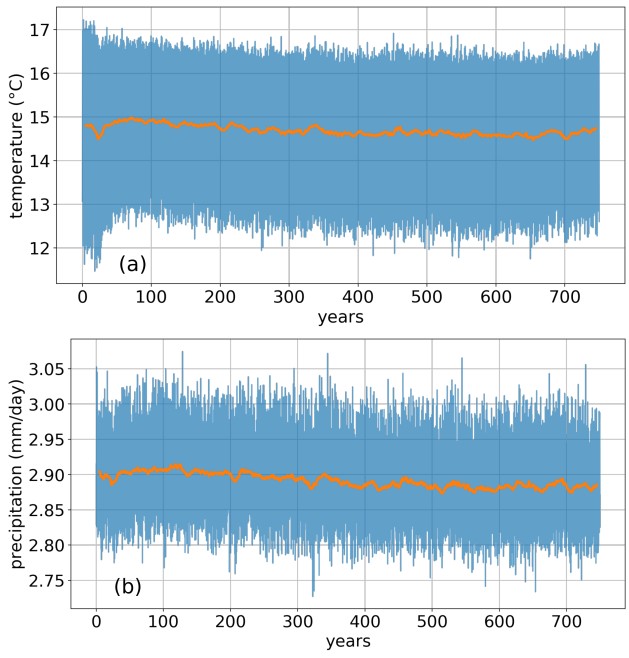

**Figure 3.** Time series of monthly mean global (a) temperature and (b) precipitation (blue lines) and the corresponding 10-year running means (orange lines) in the pi-Control experiment.

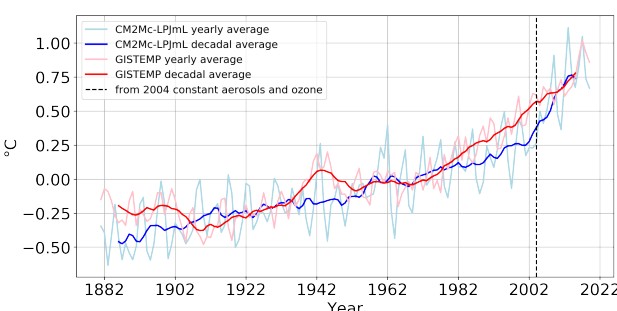

**Figure 4.** Yearly and decadal global mean temperature anomaly (relative to the reference period 1951–1980) of the TR experiment of CM2Mc-LPJmL compared to GISTEMP data from 1880–2018. Note that, from 2004 on, only greenhouse gas forcing remains, while aerosols, solar radiation and ozone are set to their corresponding 2003 values.

data (Lenssen et al., 2019). We further evaluate the spatial impact of historic climate change without land use by comparing the years 2006–2015 of the PNV experiment with the last 10 years of the pi-Control experiment in the supplement (Section S2). The temperature evolution over the historic period from 1880-2018 is well captured as compared to GISTEMP evaluation data (Fig. 4). Throughout the displayed period, temperature anomalies are negative before the year 1962 and remain positive afterwards, as climate change is accelerating. While the temperature anomalies are slightly underestimated between 1980 and



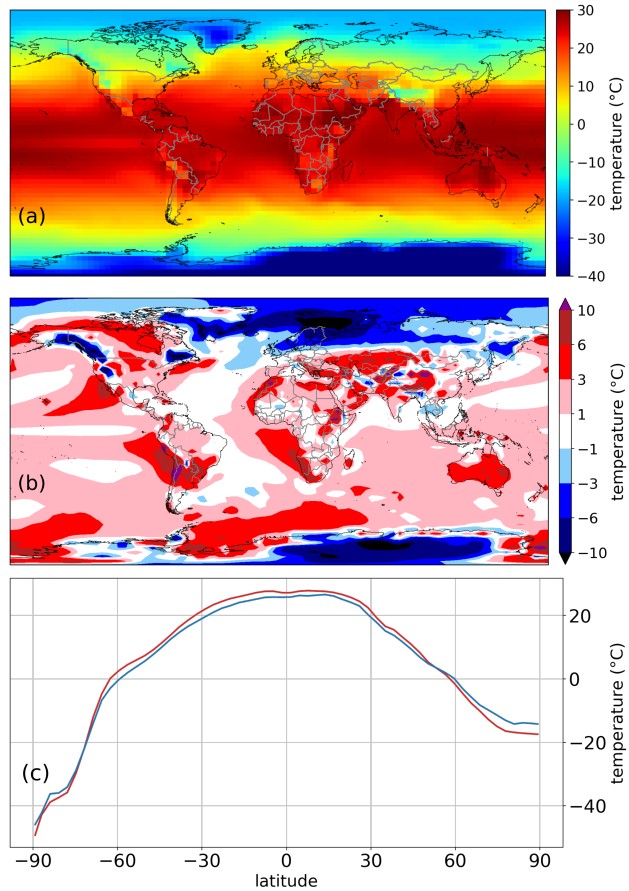

**Figure 5.** (a) Global mean surface temperature of the TR experiment over the period 1994–2003; (b) Surface temperature anomalies between CM2Mc-LPJmL (TR) and ERA5 data over the period 1994–2003; (c) latitudinal temperature mean of TR (red line) and ERA5 data (blue line) for the period 1994–2003.

2010, GISTEMP as well as the TR experiment have both an average global temperature increase of 0.75°C in the year 2018 relative to the reference period 1951–1980. Our results are also within the range of CMIP5 models (Kattsov et al., 2013; Taylor et al., 2012, Section S4). The inter-annual variability in CM2Mc-LPJmL is ca. 0.5°C and thus larger than in the GISTEMP data (ca. 0.25°C), although the decadal changes are smaller in CM2Mc-LPJmL.

    In the PNV experiment, climate change is also well captured, but weaker as compared to having included land use in the model
(Fig. S5).

### 3.1.3   Surface temperature evaluation

Basic climate patterns are well captured in the annual mean surface temperature (Fig. 5a), as temperatures are increasing from polar temperatures of below −10°C towards the equator with a maximum of ca. 25–30°C in the tropics. Desert regions are





usually warmer, while mountainous regions are colder than the surrounding area. In the high latitudes ocean cells are usually a

bit warmer than land cells, due to the ocean's ability to store heat.

Between 1994 and 2003 the average global temperature is 15.6°C compared to 14.3°C in the ERA5 data set with a NME of 0.16. While the temperatures in the tropics and temperate zone are slightly overestimated (by ca. 1°C), the poles and the boreal zone show a large negative temperature bias (up to −10°C) (Fig. 5b). The Southern Ocean has a significant positive temperature bias (ca. 3°C on average). Large differences between CM2Mc-LPJmL and ERA5 are also visible for mountainous

areas, where the temperature bias is partly due to the coarse resolution of the model, not adequately capturing the orographic influence of most mountain ranges on climate (e.g. Andes or Himalaya).

While the seasonal cycle is usually well captured in CM2Mc-LPJmL, especially in Antarctica a strong seasonal temperature bias is partly balanced out in the annual mean temperature. Temperature over Antarctica is largely overestimated during the southern-hemisphere summer, while being underestimated during the southern-hemisphere winter (Figs. S1 and S2).

The latitudinal distribution of modeled mean temperature between 1994 and 2003 (Fig. 5c) shows similar values compared to ERA5 data from high to mid-latitudes in the northern hemisphere, but a slight overestimation in parts of the temperate zone and the tropics (between 70°S and 40°N). Specifically, the cold bias in the boreal zone leads to a slight underestimation of temperature between 60°N and 90°N.

The comparison of CM2Mc-LPJmL (pi-Control) and pi-CM2Mc-LaD (as in Galbraith et al., 2011) shows that similar biases in

relation to ERA5 are present in both model versions. For example, both model versions slightly overestimate global temperature (Fig. S6). The strong regional biases as compared to ERA5 data are also present in both model setups (Fig. S6), hence not due to the implementation of LPJmL5.

### 3.1.4  Precipitation evaluation

The spatio-temporal pattern of global precipitation is well simulated with a global average of 2.86 mm/day and a maximum of

up to 10 mm/d in the tropics close to the Inter-Tropical Convergence Zone (ITCZ, Fig. 6a). Regions with little to no vegetation, such as deserts and polar areas, receive very little precipitation throughout the year.

Precipitation biases with respect to ERA5 data are, however, stronger than temperature biases with an NME of 0.50 compared to 0.16 for temperature (Fig. 6b). The biases are strongest at the equator with an apparent shift of the ITCZ. While precipitation in the Pacific is underestimated directly at the equator, it is overestimated north and south of the equator (Fig. 6b). Also northern

South America shows a large negative precipitation bias.

The seasonal patterns (Figs. S3 and S4) confirm the imprecise modeling of the ITCZ, which remains for a large part of the year north and south of the equator, while passing the equator region relatively swift. While precipitation south of the equator is overestimated, it is underestimated north of it.

The latitudinal annual mean precipitation between 1994 and 2003 (Fig. 6c) compares well with observations, displaying the

global precipitation maximum in the tropics, local minima in the subtropics, and very low values at high latitudes. The tropics, however, show a shifted maximum. While the ERA5 global precipitation maximum over the Pacific is ca. at 10°N and a local smaller maximum at -10°S, CM2Mc-LPJmL models the global maximum at roughly -10°S and a smaller local maximum at

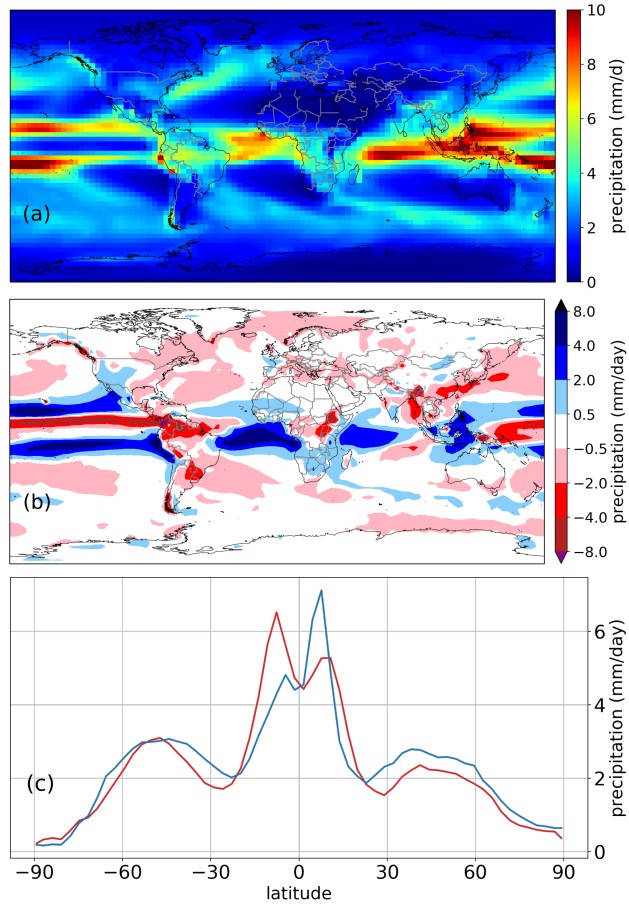

**Figure 6.** (a) Global mean precipitation of the TR experiment 1994–2003; (b) precipitation anomalies between CM2Mc-LPJmL (TR) and ERA5 data over the period 1994–2003; (c) latitudinal temperature mean of TR (red line) and ERA5 data (blue line) over the period 1994–2003.

ca. 10°N. The difference of the two maxima is less pronounced compared to ERA5.

The comparison of the results of CM2Mc-LPJmL with the original model pi-CM2Mc-LaD shows similar biases in relation to

ERA5 for both model versions. Neither of the models precisely captures the behaviour of the ITCZ, especially over the Pacific. Both models also show a large dry bias in northern South America (Fig. S6).

### 3.1.5 Vegetation cover and biomass

While the evaluation of temperature and precipitation is performed for the years 1994–2003, we compare average model results for above-ground biomass (AGB) and the dominant PFT for the years 2006–2015 due to availability of evaluation data.

Simulated AGB shows overall a good pattern, with largest values in the tropics, decreasing biomass in the subtropics and a local maximum in the temperate and boreal zone (Fig. 7b). In vegetation-free areas such as deserts or polar regions, simulated AGB



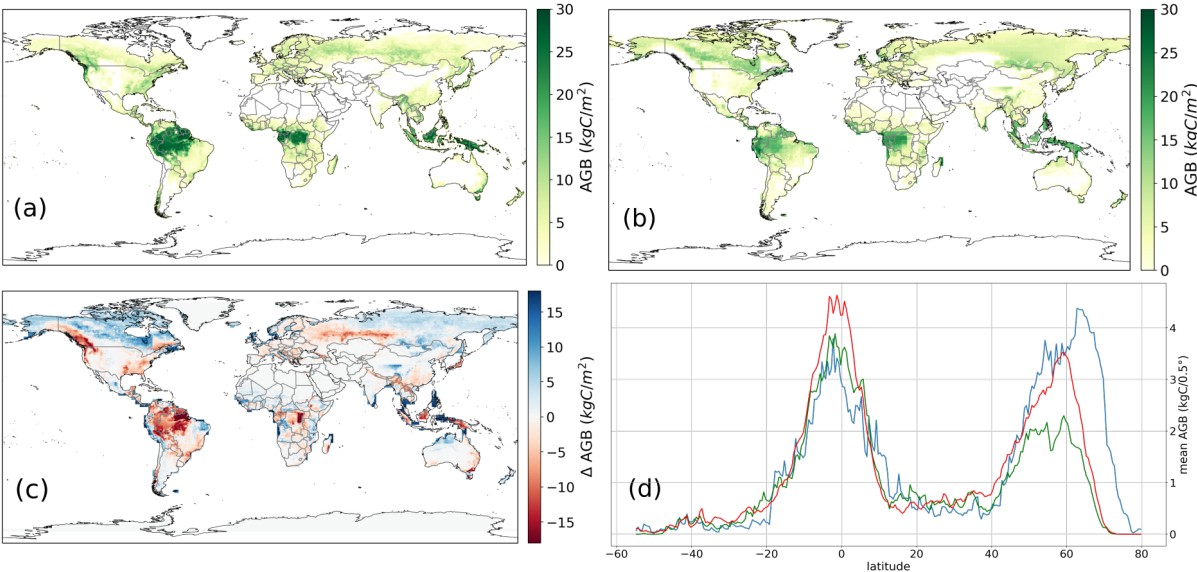

**Figure 7.** (a) Mean global above-ground biomass of GlobBiomass evaluation data. (b) Mean global above-ground biomass of CM2Mc-LPJmL (TR) over the period 2006-2015. (c) Difference of the above-ground biomass between CM2Mc-LPJmL and GlobBiomass evaluation data. Blue/red colors denote an overestimation/underestimation of biomass by CM2Mc-LPJmL. (d) Latitudinal sum of above-ground biomass from CM2Mc-LPJmL (blue line, $R^2$=0.64, NME=0.56), stand-alone LPJmL5 (green line, $R^2$=0.94, NME=0.35) input data and GlobBiomass evaluation data (red line).

is zero or very close to zero (less than $200\,\mathrm{gC/m^2}$). When comparing AGB against GlobBiomass (Fig. 7a), spatial differences emerge (Fig. 7c). While simulated AGB is slightly overestimated in boreal North America and Asia, it is underestimated in the European temperate zone and in Scandinavia, extending into eastern Europe and West-Siberia. In most of the other tem-

perate, Mediterranean-type and subtropical regions, AGB matches the observed values. In the tropics, AGB is overestimated in semi-arid regions, whereas wet-tropical rainforests are mostly underestimated, especially the eastern Amazon. AGB shows good agreement in the seasonal-dry Cerrado region in South America, but appears overestimated in the Caatinga in northeastern Brazil. In central Australia, AGB matches observations, but being overestimated in the north, and underestimated in the southeastern part of the continent (Fig. 7c).

Fig. 7d compares the latitudinal mean of CM2Mc-LPJmL and LPJmL-offline with the evaluation data. LPJmL-offline has a better performance than the coupled model with a smaller NME (0.35 vs. 0.56) and a better $R^2$ (0.94 vs. 0.64). While both models underestimate biomass in the tropics, biomass in the boreal zone is overestimated by CM2Mc-LPJmL and underestimated by stand-alone LPJmL5 compared to GlobBiomass. The LPJmL5 stand-alone version is forced by a re-analysis climatic input in a spatial resolution of 0.5° and the model is calibrated to this specific climate conditions, therefore a better model

performance is expected. Modeled biomass is also in the range of CMIP5 models (Fig. S7).

The geographic distribution of dominant PFT cover in CM2Mc-LPJmL follows the spatial pattern of the biomass distribution



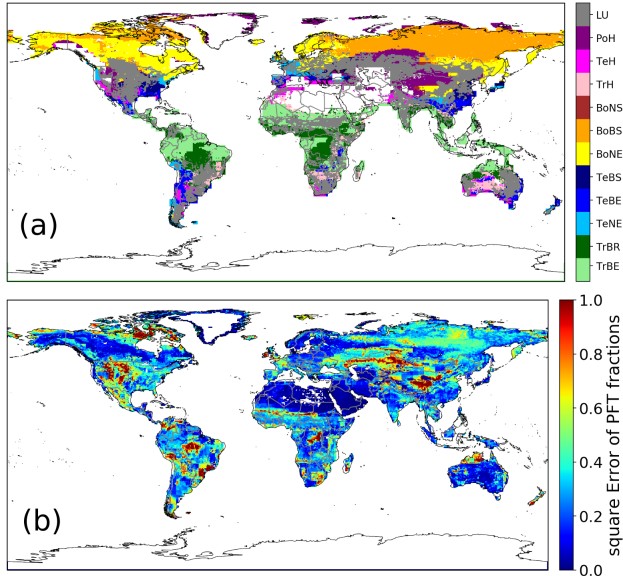

**Figure 8.** (a) Dominant PFT for each cell, modeled by CM2Mc-LPJmL. Cells with more than 50% land use are masked as grey. Cells with less than 200 gC/m$^2$ are shown white. (b) Sum of the square errors to ESAcci land cover for each PFT in each cell. Blue areas have a small error, red areas a large error. The error shown here is absolute, hence areas with a low PFT cover for both, model and evaluation data, are small compared to areas with a large PFT cover.

(Fig. 8a). The tropics are mostly dominated by the evergreen tree PFT. In the tropical savanna areas the tropical deciduous tree PFT dominates, along with the C$_4$-grass PFT. The temperate zone is dominated by land-use with some summergreen trees most common in, e.g., Europe. The boreal zone is correctly covered by boreal needle-leaved and boreal summergreen trees

and the tundra zone with polar grasses. To better visualize the model error for the PFT distribution, we produced an error map, which consists of the sum of the square error for each PFT per cell (Fig. 8b). In tropical rainforests, the error with respect to the evaluation data is relatively small. Drier savanna areas show a much larger error, as well as parts of the temperate and the boreal zone. Areas with a small FPC fraction show a small error, because the error metric takes absolute errors into account. This applies to desert regions in Africa, the Arabian peninsula and central Australia.


### 3.2 Impact of land-use changes on the coupled system

In order to isolate the impact of land-use change, we kept the climate constant and allowed land-use to change (LU-only, see Tab. 1). We compared precipitation, temperature and AGB for the years 2006–2015 of the LU-only experiment against the last 10 years of pi-Control to evaluate the absolute impact of changing land use.

Most regions with a decreasing biomass and an increasing temperature show decreasing precipitation, e.g. the Brazilian Cerrado or southern Africa. This is due to reduced evapotranspiration of agriculture and pasture compared to natural vegetation





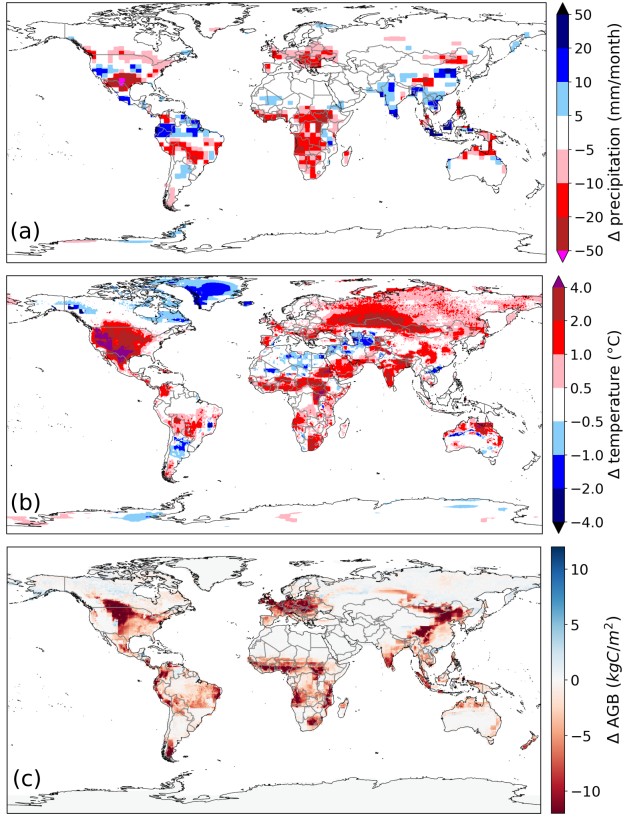

**Figure 9.** Difference between the LU-only (2006-2015) and the pi-Control (last 10 years) experiment for (a) mean precipitation, (b) mean surface air temperature, (c) mean above-ground biomass.

(Fig. 9a). Precipitation increases in regions where natural vegetation benefits from increased temperatures, for instance in mountainous regions, in India and in parts of southeast Asia (Fig. 9a).

Due to the replacement of natural vegetation by crops and managed grass, the total biomass is decreasing compared to the pi-Control experiment in regions with large land-use areas, e.g. Europe or the USA (Fig. 9c). As a consequence, surface temperature increases in these areas (Fig. 9b), leading to a global increase of ca. 0.5°C of average land-surface air temperature. In the LU-only experiment, temperature additionally increases in regions where little to no land-use change occurred, e.g. over northern Australia and Siberia (Fig. 9b). Over several sparsely vegetated areas, as in the Sahara, northeastern Canada and Greenland, temperature decreases. Temperature in tropical regions, e.g. in the Amazon basin and central Africa, are unaffected, as well as most desert and polar regions. For these regions, the amount of biomass remains the same as for the pi-Control experiment (Fig. 9 c).





## 4   Discussion

In this study we show the successful biophysical coupling of the whole-ecosystem DGVM LPJmL5 into the coarse-resolution
version of GFDL's CM2 coupled climate model (CM2Mc), replacing the simple land-surface model of CM2Mc with LPJmL5.
In order to couple the stand-alone LPJmL5 to CM2Mc, some well-functioning model elements and structures had to be revised
and modified to work in a fully coupled climate model and to meet the essential coupling variables required by the coupler
and the atmosphere modules. Even though LPJmL was developed as a stand-alone DGVM, its coupling to CM2Mc does
not significantly change the temperature and precipitation patterns, but enables us to explore biophysical climate-vegetation
feedbacks. The resulting model is furthermore in the range of CMIP5 models as stated in the Assessment Report 5 (Kattsov
et al., 2013, Fig. S7).

In Section 4.1 we discuss the challenges of coupling LPJmL5 to CM2Mc and the evaluation of the coupled system, in Section
4.2 we examine the model application to simulate historic climate and land-use change, and in Section 4.3 we present an
outlook on how the advantages of our modeling approach can be used best in future work.

### 4.1   Challenges of coupling LPJmL5 into CM2Mc

The results shown in Section 3 demonstrate that we achieved a stable model performance with respect to climate-biosphere
interactions after a potential natural vegetation spin-up period of 500 years. The climate variables temperature and precipitation
show very similar biases as CM2Mc with LaD (see Figs. 5, 6 and S6). In other words, the relatively large bias in CM2Mc in
certain regions occurs also when using the prescribed and idealized vegetation cover from LaD, and is therefore not introduced
by the coupling to LPJmL. The distribution of plant functional types and above-ground biomass are well simulated in most
regions (Figs. 7 and 8).

The performance of the coupled LPJmL5 is directly sensitive to biases in the climate input produced by the AM2 atmosphere
model. These biases can lead to a different vegetation state, which affects vegetation feedbacks to the atmosphere with possible
increasing biases in AM2. This feedback loop is responsible for the deviations in our LPJmL vegetation results compared to
stand-alone simulation experiments without such feedbacks to the atmosphere. In the latter case, an error propagation from the
climate input is avoided by forcing the model with bias-corrected climate data (Frieler et al., 2017). In our model approach we
abstained from bias or flux corrections within the coupled model to maintain more realistic feedbacks, and allow its application
to future as well as paleo-climate conditions. Furthermore, small problems in the parameterization of important processes can
lead to larger problems in the whole state of the modeled Earth system. For instance, the temperature and water cycle calcu-
lations have a strong interconnection and hence, a small error in the calculation of the water or energy cycle could lead to a
runaway temperature and cause vegetation dieback for the wrong reasons. By adapting, e.g., the calculation of evapotranspira-
tion and sublimation (see 2.3.2 and 2.3.4) we managed to keep the model relatively stable.

CM2Mc, when coupled either with LaD or LPJmL5, has a positive temperature bias of 1.3°C, which is within the range of
published Earth system models (Kattsov et al., 2013). The temperature biases in CM2Mc are especially large in the polar and
in other at least partially snow-covered regions. In the northern latitudes a negative temperature bias led to a large mortality of



vegetation in, e.g., Scandinavia in a previous model version (not shown). By adapting the simple snow model within LPJmL we obtained a stable vegetation of polar grasses and boreal trees in boreal Eurasia (see Section 2.3.4 for methods and Fig. 8 for results). A completely revised snow model or even a parallel ice sheet model could improve the modeling performance further. Globally, the biomass cover is captured well by CM2Mc-LPJmL (Fig. 7). However, in an early development version of

CM2Mc-LPJmL a dry bias in northern South America led to a strong underestimation in the biomass productivity. The modeling was improved by using the above described Penman-Monteith parameterization for evapotranspiration (Section 2.3.2) and by increasing the tropical rooting depths and hence, the soil water access of the trees (Sakschewski et al., 2020). Global biomass patterns are now also comparable with the stand-alone LPJmL5 version (Fig. 7d).

Additionally, the coarse resolution of AM2 contributes to the simulated climate and vegetation anomalies, which can be usually

expected, when running fully coupled ESMs (Galbraith et al., 2011). While LPJmL runs in the native resolution of $0.5° \times 0.5°$, the atmosphere and hence the climatic input to LPJmL, has a resolution of $3° \times 3.75°$. While this resolution is necessary for a low computational cost, it can decrease the model accuracy over, e.g., mountain ranges such as over the Andes. The model smooths the height of the Andes to the coarse grid cell size, which leads to warmer temperatures on the high mountain areas and to a colder temperature on the low areas. Small biomes, such as the Caatinga in Brazil, have the size of a few grid cells or

are even smaller than one grid cell and hence, their unique climate can not be sufficiently captured by the coarse resolution of the atmosphere model. This could be improved by using a smaller grid size, but at the drawback of larger computational costs. Since LPJmL accounts for large carbon stores, such as soil carbon, a long spin-up of several thousand years is necessary to get the carbon pools into equilibrium (Schaphoff et al., 2018a). To save computation time, this spin-up has been calculated with stand-alone LPJmL. Due to differences in the forcing of the stand-alone LPJmL version and the fully coupled model,

there is still a small offset in the beginning of the fully coupled spin-up run. After ca. 300 years, temperature and precipitation have reached a state close to an equilibrium (Fig. 3), and the model can be used for further scenarios and possible applications. Without using the multi-step spin-up, as described in the methods (Section 2.5), the time to reach a stable state would be several times larger.

## 4.2 Climate and land-use change in CM2Mc-LPJmL

In addition to regional temperature patterns, the global temperature trends in historic climate and land-use change simulations are often used as another important evaluation metric, closely related to the climate sensitivity of Earth system models (Kattsov et al., 2013). Compared to GISTEMP evaluation data (Lenssen et al., 2019), the global temperature evolution over the historic period from 1860 until 2018 is well captured in CM2Mc-LPJmL (Fig. 4). The temperature increase in this period is also comparable to Kattsov et al. (2013). Therefore the model is able to model the response of the climate system and, hence, the

response of the biosphere to historic climate change.

To realistically model regional responses to climate change, the spatial temperature biases have to be taken into account. Temperature biases on land, which are sometimes up to 2 degrees Celsius, are larger than temperature increases during historic climate change. These biases have to be considered, when interpreting results from future model runs. Furthermore, the model does not account for climate modes and extreme events (e.g. El Niño Southern Oscillation), hence the interannual variabil-





ity is smaller than expected. The interpretability of future runs is also hampered by the uncertain effect of $CO_2$ fertilization
(Clark et al., 2013; Körner, 1993). This effect is relatively strong in LPJmL, leading to an increase in vegetation productivity
at increasing $CO_2$ and temperature. The $CO_2$ fertilization effect under current climate has a stronger impact in LPJmL5 than
heat stress in a warming climate. Activating the nitrogen cycle in LPJmL5, could reduce this strong effect by taking nitrogen
limitation on vegetation productivity into account (Von Bloh et al., 2018). Historic biomass increase resulting from the $CO_2$
fertilization effect agrees, however, with previous studies (e.g. Zhu et al., 2016). A decrease in biomass in the historic period
occurs almost exclusively in regions with land-use expansion.

Land use and land use management are often neglected in Earth system models, which leads to a inaccurate modeled temper-
ature impact through land-use changes (Luyssaert et al., 2014). Since only ca. 30% of the land surface remains untouched by
humans, a correct representation of land-use practises is important for modeling climate change of the 21st century (Levis,
2010). CM2Mc-LPJmL uses the advanced land-use scheme of LPJmL5, which includes various management practises (e.g.
harvest and irrigation) for 12 different crop types.

By including land-use change in CM2Mc-LPJmL, natural vegetation is partially replaced by pasture and crops over time. This
decreases biomass which affects the climate in three different aspects: 1) Less vegetation transpires less water, which decreases
the water flux to the atmosphere, cooling by latent heat, humidity and precipitation (Gkatsopoulos, 2017), 2) the albedo of crops
is larger than that of closed forest, hence leading to a lower temperature (Unger, 2014), 3) the roughness lengths decreases,
which increases temperature (Hoffmann and Jackson, 2000). While these effects mostly consist of a cooling through larger
albedo and a warming through a smaller flux of latent and sensible heat, the net effect in CM2Mc-LPJmL is a warming climate
in most areas. Especially in the tropics the latent and sensible heat fluxes outweigh a potential cooling by albedo increases.
The biophysical effect of land-use changes is furthermore highly sensitive to changes in roughness lengths and albedo for the
different PFTs and crop functional types, as well as different management options as, for instance, a different irrigation scheme
(Kueppers et al., 2007).

Other studies, for instance, Luyssaert et al. (2014) and Alkama and Cescatti (2016) also found a warming resulting from
changes in land use and management, based on observed data. Modeling studies such as Strengers et al. (2010) and Boysen
et al. (2020) found, in contrast to our results, a cooling in temperate and boreal regions due to biophysical effects of land-use
change. While Strengers et al. (2010) used a relatively simple atmospheric model and coupling approach between biosphere
and atmosphere, Boysen et al. (2020) compared the effect of the replacement of forest with grassland for nine Earth system
models. This methodology is however different to the modeling approach in LPJmL5 where actual changes in land use and
land management are captured as well as sowing, growth and harvest of 12 different crop types, and managed grassland are
explicitly simulated.


### 4.3 Outlook

Using the advanced land use scheme of LPJmL5 and the capability of CM2Mc to accurately model climate change, the com-
bined model CM2Mc-LPJmL is a powerful tool to model future trajectories of the Earth system. It allows to calculate various


land-use change scenarios or management practises under changing climate in a computational efficient way. It is further possible to separately investigate different biophysical processes and feedbacks, while forcing the model with representative concentration pathways (RCPs). Given the speed and relatively low computational cost of the model, even long term equilibrium experiment of several hundred years can be completed within days to a few weeks.

While CM2Mc-LPJmL is fully biophysically coupled, the biogeochemical coupling is not yet included. Each submodel accounts for a local carbon cycle and balance, but the carbon cycle is not yet closed for the whole model. For this study we prescribed the atmospheric $CO_2$ concentration in all model runs and therefore a closed carbon cycle was not necessary. A fully closed carbon cycle is in the scope of future studies.

The key advantages of CM2Mc-LPJmL are the relatively fast and computational inexpensive atmosphere-ocean general circulation model (due to its relatively low spatial resolution) and the ability to investigate detailed feedbacks of the biosphere using the state-of-the-art DGVM LPJmL5. While LPJmL5 is constantly improved, recent new features such as a process-based nitrogen cycle (Von Bloh et al., 2018), a tillage system for land use (Lutz et al., 2019) or variable root growth (Sakschewski et al., 2020) can be integrated in the modelling framework consecutively and tested in the Earth system model. The coupled model also remains flexible for new model compartments such as a new atmosphere or a new ocean model, which are compatible with FMS. GFDL has already released the newest AM4 atmospheric model (Zhao et al., 2018), as well as MOM6 such as a state-of-the-art ocean model (Adcroft et al., 2019). Both could be integrated in the already existing modeling framework, and are expected to further reduce model bias.

## 5    Conclusions

In this study we demonstrate the successful biophysical coupling of the state-of-the-art DGVM LPJmL5 into the coupled climate model CM2Mc. Thereby we replace the simple static vegetation model LaD by the whole-ecosystem model LPJmL5. To achieve this goal, major adaptations were implemented in LPJmL5. These included the implementation of a new canopy module and a sub-daily time step in LPJmL5. The performance of the newly coupled model is similar to CM2Mc-LaD (Galbraith et al., 2011) and comparable to CMIP5 (Kattsov et al., 2013). The NME of temperature and precipitation showed good values of 0.16 and 0.50. The vegetation cover and biomass (NME=0.56) is also well captured compared to evaluation data. Some regions, however, exhibit large temperature and precipitation biases due to the old atmosphere and its coarse spatial resolution. The model shows furthermore a stable performance over 750 years and reasonable reactions to climate and land-use change. The average surface temperature increases by ca. 0.75°C in 2018 compared to 1950–1980. Land-use expansion over the last 300 years led to a generally drier and ca. 0.5°C warmer climate.

The fully coupled energy and water cycle allows investigating the impact of biophysical atmosphere-biosphere feedbacks on global climate trajectories and quantifying impacts of deforestation or afforestation scenarios. CM2Mc-LPJmL might further help in identifying tipping points and planetary boundaries especially in the biosphere. By using LPJmL5 we can make, e.g., use of its advanced land use scheme, the sophisticated process-based fire model SPITFIRE (Thonicke et al., 2010), a representation of permafrost and a state-of-the-art water cycling (Schaphoff et al., 2018a) and incorporate future model developments.

*Code and data availability.* MOM5 code and example configurations are public available via the project homepage[3]. Further information about the CM2Mc setup and BLING is available at the Integrated Earth System Dynamics Laboratory[4]. Code, input data, and model output of the coupled system are stored in PIK's long-term archive and will be made available to interested parties upon request. It is planned to
publish the coupling interface and the used LPJmL version as open source with the publication of this article.

*Author contributions.* MD, KT, GF, BS, WvB, SP designed the research with input from WH and MF. WvB and SP developed the technical framework for the interface between FMS and LPJmL with input from MD and SS. MD, WvB, SS and SP developed equations for the water and energy cycle for the coupling interface which are not present in stand-alone LPJmL. MD conducted the simulations and prepared the figures. MD prepared the manuscript with input and feedback from all co-authors.

*Competing interests.* The authors declare that they have no conflict of interest.

*Acknowledgements.* This paper was developed within the scope of the IRTG 1740/TRP 2015/50122-0, funded by the DFG/FAPESP (MD and KT). KT and BS acknowledge funding from the BMBF- and Belmont Forum-funded project "CLIMAX: Climate Services Through Knowledge Co-Production: A Euro-South American Initiative For Strengthening Societal Adaptation Response to Extreme Events", Grant no. 01LP1610A. The authors gratefully acknowledge the European Regional Development Fund (ERDF), the German Federal Ministry of
Education and Research and the Land Brandenburg for supporting this project by providing resources on the high performance computer system at the Potsdam Institute for Climate Impact Research. Thanks to Erik Gengel, who worked towards this coupling in his Master Thesis. The acknowledge the World Climate Research Programme's Working Group on Coupled Modelling, which is responsible for CMIP, and we thank the climate modeling groups (listed in Table S1) for producing and making available their model output. For CMIP the U.S. Department of Energy's Program for Climate Model Diagnosis and Intercomparison provides coordinating support and led development of
software infrastructure in partnership with the Global Organization for Earth System Science Portals.

---

[3]https://mom-ocean.github.io/
[4]https://earthsystemdynamics.org/models/bling/



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
