# Peer review of "CM2Mc-LPJmL v1.0: Biophysical coupling of a process-based dynamic vegetation model with managed land to a general circulation model"

_Geoscientific Model Development, 2020_

## Referee Comment (RC3)

**Review GMD-2020-436**

Dear authors

The topics in the manuscript "CM2Mc-LPJmL v1.0: Biophysical coupling of a process-based dynamic vegetation model with managed land to a general circulation model" could be interesting for the readers in GMD and a central topic in this special issue "The Lund-Potsdam-Jena managed Land (LPJmL) dynamic global vegetation, hydrology and crop model - developments, evaluations and documentation". Authors tried to describe a new earth system model based on climate model CM2Mc coupled with LPJmL. The simulation outputs by the CM2Mc-LPJmL seemed to nicely agree with the previous simulation results in both historical and future period. The current draft is readable. However, I feel that it is not sufficient in describing the model. Therefore, I did not make an acceptance decision in the current draft of the version.

**General comments**

The major comments are as follows;

1. I understand that creating a new ESM is challenging, but I did not understand what was new about this new ESM compared to the existing ESM. For example, it describes forest fire and permafrost schemes, but it is not clear whether these are new to the existing ESM. At least, there are no results showing any changes due to the introduction of these schemes into the simulation. The significance of new coupling DGVM with an atmospheric circulation model rather than extending the existing "ESM" is also unclear. It would be nice to have more explanation on this point.

2. Honestly, I don't know how much description is appropriate in describing such a huge model. Therefore, this is just a hunch, but I felt that the description of each elemental model was still too small. For example, as a person who deals purely with GCMs, is the description of biogeochemical models sufficient? At least, I can hardly imagine the contents of AM2 model and MOM5 models.

**Individual comments**

**L89–94** It seemed that the limited variables are interconnected

**L89–94** I'm not familiar with ESMs. So, for me, it is hard to imagine how a coupler work between models. How do you reconcile the conservation of energy and mass as variables are exchanged between models with different resolutions?

**2.1.2 & 2.1.3** Please don't use abbreviations in the section title. Like 2.1.1 or more insightful title is better.

**2.2** Same above.

**L105** What are the input variables for this component. Please specify them as in L118.

**L110** dynamic core of ?

**L111** Sorry, what is a C and D grid. Perhaps, it is a very common term in the climate model study field.

**L112** What do the tracers mean? What do the dynamics suggests?

**L121** LPJmL5 -> LPJmL ver. 5? Please add the citation for this version.

**L126** Please summarize all plant functional types in the manuscript or the supplemental material. Although the simulation results are presented in Fig 8, the readers can only reach the abbreviation about PFT.

**L131** Which dataset was used for the prescribed land-use input?

**2.3.2** Section 2.3.2 should be incorporated into Section 2.2 (LPJmL part).

**L174–179** It is better to describe this sentence in the introduction.

**L321** Please add the citation for the "historic land-use data from 1700".

**L354** Please provide here which variables $y_i$ are evaluated by this metric.

**L367: 3.1.1** How about the stability in other variables in other model components such as PFT distribution, C stock in terrestrial and Ocean ecosystems?

**L410** Are there any improvement in the CM2Mc-LPJmL output that should be mentioned, compared to CM2Mc-LaD?

**L414 & Figure 6** I don't know the manner of climate science, but why do you compare annual precipitation in mm/day instead of mm/year? It is hard to intuitively understand the size of the bias.

**L450–** There are no information about PFTs in detail. I can see only abbreviation in the legend of Figure 8. So, please add this information in material and method or in appropriate place.

**L485** If the results of the comparison with CMIP5 are important, it would be better to include them as a figure in the draft instead of sending them to the supplement.

**L587–590** Some of this text could be taken to the intro and explained as to why we are creating a new ESM using the coupler.

**Figure 1** Figure 1 is not referred in the text. Are there any exchange between MOM and LPJmL? How about atmospheric pressure between AM2 and LPJmL.

---

## Author Comment (AC2)

**Reviewer 1**

*We thank reviewer 1 for the detailed and thorough comments. Our replies to the comments are inserted below in blue colour.*

Dear authors,

after reading your paper, I had generally good impressions, but I also found the methods section confusing.

The overall structure of the paper is fine. I really like the choice of experiments for the model evaluation, and it provides a clear overview of the strengths and limitations of the model. I do think this model has potential to help advance our understanding of climate-land surface-human activity interactions (high scientific significance), and therefore think that this work has good scientific value.

However, I would recommend a thorough revision/restructuring of the methods section before publishing. I found it quite confusing to work out how exactly the model calculates certain things on a first reading, and a second reading (with pen and paper) still left things unclear. Because of this I couldn't rate the presentation and the reproducibility high. I clicked "major revisions", but the focus is mainly on the explanation of the methods rather than the science. I am of course willing to review the revised manuscript.

We thank the Reviewer for the generally positive review. We are also very grateful for the thorough analysis of our methods, which led to a great improvement of the presentation of the model description.

One important part to emphasize is the role of the FMS coupler. FMS standardizes the interfaces between various model components and handles the fluxes between them. Hence, FMS has several tasks of a LSM, including the calculation of aerodynamic resistance, drag, and air stability calculation by taking into account information from the land model and the atmosphere. In our coupling approach we just provide the same variables to FMS as Lad, which are roughness length, albedo, humidity and surface/canopy temperature. The coupler then calculates the fluxes to the atmosphere, e.g. q_flux, and makes this information available for the land model in the next time step. Since our coupling approach is limited to the interface between LPJmL and FMS, it is beyond the scope of this paper to explain in detail the processes within FMS and the atmosphere. To avoid misunderstandings, we also changed the name of the blue box of Fig. 2 in the paper to FMS coupler. The variables stated there are all calculated in the coupler, using input from the atmosphere model. To clarify and explain better the role of the coupler, we added to section 2.3.

*"In this Section we describe our coupling approach at the interface between the land model (LPJmL) and the FMS coupler. FMS calculates the fluxes between the different model components and provides this information to the sub-components. The tasks of the coupler also include the calculation of air stability and surface drag, hence it has some functionality of a land surface model. Because it is beyond the scope of this paper to explain the processes within FMS in detail, we refer to Milly and Shmakin, 2002 and Anderson et al., 2004 for further details."*

In the following we answer the comments and questions of the Reviewer in detail:
* * *
It is stated in line 139 that the variables that are exchanged on the "fast" time step are: canopy humidity, soil temperature, canopy temperature, roughness length and albedo. Some first questions that come to mind:

- Does canopy temperature refer to bulk surface temperature?

The reviewer is right, that by using an energy balance equation we are calculating a surface temperature. Since we do not have a height dependent scheme in our canopy, this surface temperature is used for the calculation of the canopy evapotranspiration and humidity. Hence, we effectively use the calculated surface temperature as the temperature of the canopy layer, similar to the parameterization in LaD. We added this information to 2.3.2 to express this more clearly:

*"Since our approach does not account for a height dependent canopy temperature, we used here the surface temperature as an approximation for the canopy temperature, which is then used to calculate canopy humidity and evapotranspiration. Hence, surface temperature and canopy temperature are assumed to be the same, following the approach in the LaD model (Milly et al. 2002)."*

- Do all the tiles in a gridcell have the same "canopy" temperature (even the bare soil fraction?)

In LPJmL, each grid cell contains several so-called stands, one for natural vegetation, and one for each crop type (CFT) and for managed grassland. The natural vegetation is subdivided into plant functional types (PFTs). The surface/canopy temperature is calculated separately for each stand, and then averaged over all stands before it is used for humidity calculations and passed to the coupler. We added an explanation of the stand structure of LPJmL to the LPJmL model description in 2.2  (see our response to your last question), and added the following sentence to 2.3.2:

*"While the temperature is calculated individually for each stand, a weighted average over all stands within one gridcell is used in the humidity calculation and passed to the coupler."*

- What is soil temperature and why does the atmosphere need it?

In our implementation, the surface/canopy temperature interacts with the atmosphere. However, for simulating permafrost, soil hydrological processes and soil heat transfer, soil temperature is computed within LPJmL5. Therefore, soil temperature is the temperature in

the various soil layers in LPJmL5 (i.e. 6 layers in the current version). It is not needed by the coupler or the atmosphere, just internally used by LPJmL. Since the climate input from the coupler and the whole temperature routine operates on the fast time-step, the soil temperature is also calculated hourly. For the other daily calculations in LPJmL5 a daily mean is applied. The soil temperature uses the air temperature as an input, which is highly dependent on the new canopy temperature. We also added the description of the time step for both temperature calculations (fast time step). To clarify, we added in Section 2.3.2:

*"The soil temperature is still important for internal processes in LPJmL5 such as permafrost but not needed in the coupler to calculate fluxes from the land to the atmosphere. The calculation of heat transfer in the soil layers uses the heat-convection scheme as in stand-alone LPJmL5 (Schaphoff et al. 2018a) by taking into account the air temperature, which highly depends on the canopy temperature. Both temperature calculations, for the surface/canopy temperature and for the soil temperature, operate on the fast time step."*

- How does the atmosphere use roughness length to calculate the turbulent fluxes? Does it assume neutral stability conditions?

The aim of our paper is to describe the interface between the land component (LPJmL5 in our case) and FMS coupler. With respect to the roughness length, we have not modified any other parts in the coupler nor the atmosphere. Therefore, any handling of roughness length and assumptions referring to stability conditions in the atmosphere remain the same. LPJmL5 provides roughness length, which depends on the FPC of the PFT composition as well as CFT assemblage present in any grid cell. The coupler uses this and additional information from the land model and the atmosphere to calculate (and exchanges with the land component) aerodynamic resistance and surface drag, which depend on the roughness length, the height, where potential temperature is defined, and on the Monin-Obukhov length. In the coupled setup of the CM2Mc model, the atmosphere is not set to neutral stability but dynamically computes stability for each grid cell and throughout the sub-daily cycle. More information is available in Milly et al. 2002 and Anderson et al. 2004. We added a better explanation about the calculations in the coupler to 2.3 (see our response to first general remark) and added a sentence on the purpose of the roughness length to 2.3.3:

*"The coupler uses the roughness length to calculate aerodynamic resistance and surface drag and provides these variables to the different submodels of the ESM."*

EVAPOTRANSPIRATION AND CANOPY HUMIDITY CALCULATION

The aim of this calculation is, given q_flux and e_a, from the previous time step, calculate updated q_ca, which will be used by the atmosphere (along with roughness length) to calculate an updated q_flux and e_a, and so on.

Yes, this is one important aim, but please note that the calculation of q_flux is performed within the FMS coupler. Both the land model and the atmosphere provide their boundary humidity conditions. The coupler then calculates the moisture flux between atmosphere and land, q_flux, and provides it to the land model as well as to the atmosphere. Other aims of

this calculation are to provide essential variables for calculations done in LPJmL such as evapotranspiration for plants or for calculating the energy balance.

Since the atmosphere provides q_flux, this means it must be assuming neutral stability conditions or using some parametrized stability functions (e.g. Louis 1979) to avoid the need for the LSM to iteratively calculate q_flux. How does this happen?

q_flux is provided and calculated by the FMS coupler by taking into account the humidity from the land model and from the lowest layer of the atmosphere. Hence, the coupler provides some necessary LSM functionality. We take q_flux to compute delta_q_ca (equation 9) to couple ET, as simulated by LPJmL5, and relate it to the canopy/surface temperature.

As explained above, the atmosphere does not assume neutral stability. However, the explanation of how exactly the coupler interacts with the atmosphere is beyond the scope of this paper. Since we did not modify any of those functions, we focus on the interaction between land and coupler in our paper. We added a better description of the processes within the coupler to Section 2.3 (see response to first general remark). For further information on the interaction between coupler and atmosphere we refer to the respective publications and model documentations which we also cite in the paper (Milly et al. 2002, Anderson et al. 2004, Galbraith et al. 2011).

The algorithm proceeds as follows

1. - Calculation of potential evapotranspiration. According to Eq. 1, you are using a daily value of ET0. Why is this? Are you calculating this quantity on a daily or a subdaily basis?

The implementation of ET0 is subdaily. While the subdaily time step is currently set to one hour, it is however flexible and hence, we stated here the more general daily form of the Penman Monteith equation. For the implementation the resulting evapotranspiration is then divided by the number of timesteps per day, which is however not necessary in the model description, in our opinion. We added the information in the text in 2.3.2 and in a new overview figure (see below), that ET0 is calculated on a subdaily basis:

*"ET_0 is presented here in the general daily form, but applied to the model on the subdaily timescale, therefore divided by the number of time steps per day (in the current version 24)."*

2. - Potential canopy conductance from assimilation using Medlyn's model. Is assimilation daily or subdaily? Assimilation, like stomatal conductance, varies greatly during the diurnal cycle.

This step is unfortunately calculated daily, because it is a core calculation in LPJmL5 in relation to the photosynthesis routine. By changing this to a new subdaily parameterization, large parts of LPJmL would have to be adapted as well and the results would deviate largely from the original model. Since we wanted to keep the basic functions of LPJmL5, e.g. photosynthesis, as close as possible to the stand alone version, we decided to use daily values in this step. The actual water-stressed canopy conductance is, however, calculated

subdaily using the Penman-Monteith formula (Eq. 1 in the manuscript). We added to Section 2.3.2:

*"While the new potential evapotranspiration is calculated in the subdaily time step, the non-water-stressed canopy conductance is calculated in a daily time step, due to the daily calculation of the photosynthesis in LPJmL5."*

3. - Supply/demand approach to calculate TRANSPIRATION, from which an stressed canopy conductance is derived using Penmann-Monteith again. -> OK.

I am guessing all these steps occur at a subdaily timestep but this is not clear from the text, especially because of the daily potential evapotranspiration (Eq. 1).

Almost all steps are subdaily, only potential canopy conductance is computed daily (see response to point 2, above). The variables $g_i$ and $g_e$ are also calculated daily, but they are based on a simple approach and do not have any subdaily input variables. We added an explanation about the length of the corresponding time step in Section 2.3.2 (see above) and information about the time step for each step of the calculations to the text and added a new overview figure for the humidity calculation (see new overview figure below). By this, we hope to have clarified at which time step those variables are calculated.

4. - Calculation of parametrized soil evaporation conductance -> OK

5. - Calculation of parametrized interception conductance -> What is Pr in Eq. (5)? Why are you using equilibrium evapotranspiration (Eq) in this formula, rather than the new Penmann-Monteith based ET0?

We thank the reviewer for noting an error in the formula. Indeed, we used ET0 instead of $E_q$ in the model and corrected Eq. 5 in the paper accordingly. Pr denotes the precipitation, whose variable declaration was added in 2.3.2. Equation 5 now reads:

$$g_i = \text{GI}_{\text{MAX}} \cdot i \cdot Pr/ET_0 \cdot f_v$$

Now we get to the updating of Dq_ca (I use D for the Delta symbol).

6. - Where does Eq. (9) come from? If there is a reference for this equation, you should mention the principle from which it is derived and give the reference (e.g., q_ca is calculated from water conservation as in Author et al. (year).). If you have derived this equation yourself, I would like to see it derived in the paper too, maybe in an appendix if you don't want to put it in the main text. Apart from the derivation, a clear explanation of where or how each of the terms is calculated would be welcome (for example, the evaporation-humidity gradient).

We thank the reviewer for identifying this gap in the model description. In fact, the derivation of Eq. (9) is not straightforward since it was not documented explicitly in the original LaD paper. Using earlier publications (Milly et al., 1991) we have now derived Eq. (9) with the following steps:

Assuming equilibrium conditions the flux entering the canopy layer from soil and vegetation through evapotranspiration ET or $E_{in}$ equals the flux leaving the canopy layer into the atmosphere $q_{flux}$ or $E_{out}$.

$$E_{in}(t) = E_{out}(t) \tag{C1}$$

The water fluxes for the next time step t+1 yield:

$$E_{in}(t) + \frac{dE_{in}}{dt} = E_{out}(t) + \frac{dE_{out}}{dt}, \tag{C2}$$

using

$$E(t+1) = E(t) + \frac{dE}{dt}. \tag{C3}$$

Using (Milly and Shmakin, 2002) and Eq. 7 from this paper yields for E:

$$E = \frac{\rho}{r_a}[q_s - q_a] = g_c[q_s - q_a], \tag{C4}$$

where $\rho$ is the air density, $r_a$ the aerodynamic resistance, $g_c$ the canopy conductance, $q_s$ the saturation humidity and $q_a$ the actual humidity. The derivation of Eq. C4 can be used for $\frac{dE_{in}}{dt}$. Eq. C2 then yields:

$$\frac{dE_{out}}{dt} = E_{in} - E_{out} + g_c\frac{d[q_s - q_a]}{dt} \tag{C5}$$

Rearranging this equation yields:

$$\frac{dE_{out}}{dt} + \frac{dq_a}{dt} \cdot g_c = E_{in} - E_{out} + \frac{dq_s}{dt} \cdot g_c \tag{C6}$$

Expanding $\frac{dE_{out}}{dt}$ with $q_a$ yields:

$$\frac{dq_a}{dt} \cdot \frac{dE_{out}}{dq_a} + \frac{dq_a}{dt} \cdot g_c = E_{in} - E_{out} + \frac{dq_s}{dt} \cdot g_c \tag{C7}$$

Rearranging Eq. C7 yields:

$$\frac{dq_a}{dt} = \frac{E_{in} - E_{out} + \frac{dq_s}{dt} \cdot g_c}{\frac{dE_{out}}{dq_a} + \frac{dq_a}{dt} \cdot g_c} \tag{C8}$$

Expanding $\frac{dq_s}{dt}$ with dT for the temperature change yields:

$$\frac{dq_a}{dt} = \frac{E_{in} - E_{out} + \frac{dq_s}{dT} \cdot \frac{dT}{dt} \cdot g_c}{\frac{dE_{out}}{dq_a} + g_c}, \tag{C9}$$

which is the final form for the change of actual humidity over a timestep. By using $\Delta q_{ca}$ for $\frac{dq_a}{dt}$, ET for $E_{in}$, $q_{flux}$ for $E_{out}$ and $\frac{de}{dq}$ for $\frac{dE_{out}}{dq_a}$ the final form yields:

$$\Delta q_{ca} = \frac{ET - q_{flux} + \frac{dq_s}{dT} \cdot g_c \cdot \frac{dT}{dt}}{\frac{de}{dq} + \cdot g_c}. \tag{C10}$$

We added this derivation to the Appendix C. The gradients in the equation over time are the differences over one time step, while the gradient of q_sat over temperature is the slope of the water vapor pressure curve in relation to temperature.

7. - ET in Eq. (9) is the total evapotranspiration, coming from all the gridcell tiles, i.e., using the g_c, g_i and g_e calculated above. How do you arrive to ET from gc? I guess you use Penmann-Monteith again, but this is not specified in the text.

Yes, we used the Penman-Monteith equation as in (1), now applying the water-stressed canopy conductance gc. We thank the Reviewer for noting this and added the missing information to the explanation of this step of the methods in 2.3.2:

*"...and ET the final evapotranspiration, consisting of transpiration, evaporation, interception and sublimation from surface or vegetation into the canopy layer. For the calculation of ET we used the Penman-Monteith equation (Eq. 1), now applying the total water-stressed canopy conductance g_c (Eq. 7)."*

SURFACE ENERGY BALANCE

- Line 245: K and L: incoming, outgoing or net? If it is incoming it cannot be net. Same for outgoing. Net means incoming minus outgoing. Also, if one is incoming and the other one outgoing, shouldn't they contribute in opposite ways, and thus have different signs? I suspect K+L is just net radiation. In the reference you give for this equation (Milly and Schmakin 2002) the radiation balance in that paper looks different. Maybe you could simply use Rn in Eq. (10), and mention that net radiation is calculated as in Milly and Schmakin (2002)? Also, how do these two relate to the net radiation, Rg, used in Eq. (1)?

We use the net incoming shortwave radiation. This means, compared to the incoming radiation at the top of the atmosphere, the radiation which reaches the surface after part of the radiation is absorbed by the atmosphere or reflected by clouds or surface. The FMS coupler provides shortwave downward radiation, reaching the canopy. Using the surface albedo from LPJmL5 we then calculate the net shortwave radiation. The net outgoing long-wave radiation also takes into account incoming long-wave radiation through the greenhouse gas effect. Hence, these are not net total radiative fluxes, but the net of incoming and outgoing fluxes. We agree with the Reviewer, that it is easier to use just the total net radiation at the surface, as it has been used in equation 15 in Milly and Schmakin, 2002. It is also true that we already used the net radiation R_n in Eq. (1) and use the same variable now in the revised manuscript also for Eq. 10:

$$\Delta T = \frac{R_n - m \cdot LE_f + ET \cdot LE_v - Q_{sn} - H}{C_s \cdot \Delta_t},$$

- Line 248: In this implementation, the boundary temperature to the soil layers and the canopy temperature are the same as in LaD (Anderson et al., 2004). -> Is there a comma missing here? Are the boundary temperature between the soil layers and the canopy temperature the same, as in LaD? Or are they the same as in LaD? So I guess what this means is that the soil temperature calculation is now driven by the bulk surface temperature rather than the air temperature. Do tiles in LPJmL have separate soil columns or do they all share the same soil column?

We thank the Reviewer for noting this. There is indeed a comma missing. In the revised manuscript we deleted this sentence but added the following information to 2.3.2 (as stated above):

*"Since our approach does not account for a height dependent canopy temperature, we used here the surface temperature as an approximation for the canopy temperature, which is needed for the calculation of canopy humidity and evapotranspiration. Hence, surface temperature and canopy temperature are assumed the same, following the approach in the LaD model (Milly et al. 2002)."*

The soil temperature calculation is still depending on the air temperature, provided by the coupler, which is of course very similar to the surface/canopy temperature. As explained above, the surface temperature and the canopy temperature are assumed the same in our implementation, which could be improved in further development of the model. All stands in LPJmL have their own surface temperature, which is later averaged before provided to the coupler. As stated above we added an explanation for this to 2.3.2:

*"While the temperature is calculated individually for each stand, a weighted average over all stands within one gridcell is used for the coupler and the calculation of the humidity."*

- Again, reading the LaD reference, it is clear that they do calculate fluxes taking into account air stability, while it is not clear how you do this in your model. It is normally the job of the LSM to calculate the turbulent fluxes. I guess this is now done in some sort of surface layer in the atmosphere model. But it should be clear why you can replace Lad with LPJmL and avoid the stability calculation.

As discussed in the beginning of this response letter, the calculation of the fluxes between land and atmosphere, as well as air stability are done in the FMS coupler. By replacing LaD we just have to provide the necessary variables (roughness, humidity, temperature and albedo) and do not change the handling of the fluxes within the coupler. Hence, it is not necessary to calculate the stability calculations in LPJmL. For more details, see our response at the beginning of the response letter to your general remark.
* * *
***I want to reiterate that I find the scientify quality of the paper good.*** However, I think all the points above should be addressed in order to make it clearer to the reader how the model works, especially since this is a model description paper. This would substantially improve clarity and reproducibility. Special attention should be given to:

- The air stability question.

That is handled by the coupler in cooperation with the atmosphere component, see remark about our text improvement above.

- Derivation of Eq. (9).

- The confusing daily/subdaily issue in Eq. (1). If ET0 is subdaily, the equation needs a correction. If it is daily, it needs justification, given that it is used to calculate $g_c$, which varies diurnally.

Again, we are very grateful for the positive evaluation of the scientific quality and for the thorough evaluation of the model description and for noting a few inconsistencies. We hope to have now clarified the issues with our explanation in our responses above and our modifications in the manuscript text.

Some further suggestions to improve the exposition, apart from addressing the above points:

- Figure 2 is a bit cluttered. Maybe you could add a similar figure besides it where you explain the logic of the evapotranspiration scheme.

This is a very good idea. The new version of the manuscript now includes a flowchart for the logic of the evapotranspiration/humidity scheme.

[Figure]

Fig. 1: Schematic overview of the most important processes to determine the canopy humidity. The yellow color denotes newly implemented processes in the new canopy layer in LPJmL5, green internal LPJmL5 calculations and blue denotes input, provided by the FMS coupler. Daily processes are indicated by a dotted line, processes operating on the sub-daily time step by a solid line.

- A table of symbols would be very helpful. You could list the symbol, the units, whether it is an input to the LSM or an output to the atmosphere and the time step at which it is calculated/exchanged.

We added the table for the variables and parameters to the Appendix A of the revised manuscript. Further information about time steps and input/output to the coupler/atmosphere can be found in the schematic figures, including the new one from the remark above.
* * *
There are also a few minor points:

- Line 435: "Simulated AGB shows overall a good pattern, with largest values in the tropics, decreasing biomass in the subtropics and a local maximum in the temperate and boreal zone (Fig. 7b)." I think this is more easily seen in Fig. 7d than in Fig. 7b.

We thank the Reviewer for noting this and changed it to Fig. 7d.

- The DOI for Shapoff et al 2018a is wrong (it takes you to the second part of the paper instead of the first one) (https://doi.org/10.5194/gmd-11-1343-2018)

We thank the Reviewer for noting this and changed the DOI to the first paper, Schaphoff et al. 2018a.

- The lines in plot 7d can be confusing for a colorblind person. I would suggest changing either the green or the red one, and also making the lines a bit thicker.

Sorry for not thinking of colour blindness in the first place. We thank the Reviewer for the suggestion and changed the green line to a black one and slightly increased line thickness.

[Figure]

Fig 2.: Latitudinal sum of above-ground biomass from CM2Mc-LPJmL

I have another minor comment/suggestion that I think would make things clearer. It would be helpful to have a short description of how LPJmL represents land in section 2.2. Two or three lines would suffice. Something like this paragraph from section 2 of von Bloh et al. (2018):

"In the LPJmL model vegetation is represented by different plant functional types (PFTs) that can establish concurrently within a cell. These established PFTs share the same soilstand and compete for light, water, and nitrogen resources,while crop functional types (CFTs) are established exclusively at sowing on their own soil stand."

This would answer several questions that can come to mind while reading your manuscript without having to go to the more complete descriptions.

We thank the Reviewer for this useful suggestion. We hope our earlier answers were clear about the individual calculation of temperature in the stands within one cell. Following the suggestion of the Reviewer we added a sentence to the model overview of LPJmL in section 2.2. :

*"LPJmL5 simulates global vegetation distribution as the fractional coverage (foliage projective cover or FPC) of plant functional types (PFTs, Appendix B) which changes depending on climate constraints and plant performance (establishment, growth, mortality). Plants establish according to their bioclimatic limits (adaptation to local climate) and survive depending on their productivity and growth, their sensitivity to heat damage, light and water limitation as well as fire-related mortality. The interaction of these processes describes the simulated vegetation dynamics in natural vegetation. The model also simulates land use, i.e. the sowing,growth and harvest of 14 crop functional types and managed grassland (Rolinski et al., 2018). The proportion of potential natural vegetation and land-use within one grid cell is determined by the prescribed land-use input. Each type of land cover, i.e. natural vegetation, managed grassland or crops, have their own respective stand. While receiving the same climate information, soil and water properties as well as carbon-related processes are simulated separately. "*

---

## Author Comment (AC3)

**Reviewer 2**

*We thank reviewer 2 for the detailed and thorough comments. Our replies to the comments are inserted below in blue colour.*

Dear authors
The topics in the manuscript "CM2Mc-LPJmL v1.0: Biophysical coupling of a process-based dynamic vegetation model with managed land to a general circulation model" could be interesting for the readers in GMD and a central topic in this special issue "The Lund-Potsdam-Jena managed Land (LPJmL) dynamic global vegetation, hydrology and crop model - developments, evaluations and documentation". Authors tried to describe a new earth system model based on climate model CM2Mc coupled with LPJmL. The simulation outputs by the CM2Mc-LPJmL seemed to nicely agree with the previous simulation results in both historical and future period. The current draft is readable. However, I feel that it is not sufficient in describing the model. Therefore, I did not make an acceptance decision in the current draft of the version.

The major comments are as follows;
1. I understand that creating a new ESM is challenging, but I did not understand what was new about this new ESM compared to the existing ESM. For example, it describes forest fire and permafrost schemes, but it is not clear whether these are new to the existing ESM. At least, there are no results showing any changes due to the introduction of these schemes into the simulation. The significance of new coupling DGVM with an atmospheric circulation model rather than extending the existing "ESM" is also unclear. It would be nice to have more explanation on this point.

2. Honestly, I don't know how much description is appropriate in describing such a huge model. Therefore, this is just a hunch, but I felt that the description of each elemental model was still too small. For example, as a person who deals purely with GCMs, is the description of biogeochemical models sufficient? At least, I can hardly imagine the contents of AM2 model and MOM5 models.

An important goal of our work is to couple the state-of-the-art DVGM LPJmL to a reasonably fast atmosphere-ocean model. This opens up research possibilities involving the interaction of and feedback between climate system and biosphere, including land use, which are not possible with either a simpler land model, nor with stand-alone LPJmL. Using a coarse-grid atmosphere-ocean setup enables simulations on centennial to millennial time scales, and/or ensemble runs.
The aim of this paper is not to describe a completely new ESM but the coupling of existing parts. We used the existing ESM CM2Mc from GFDL and replaced the land surface module LaD with LPJmL. Hence, in our opinion, it was not necessary to describe all parts of the ESM, which are individually published and described in this configuration in Galbraith et al. 2011. Instead, we focus on the coupling between FMS and LPJmL5 and the changes implemented in LPJmL5. In the introduction and discussion we stated the advantages of LPJmL5 compared to the standard land model component used in CM2Mc, which include

dynamic vegetation, process-based fire, permafrost and an advanced land use scheme. While we are showing the impact of land use in greater detail it is not within the scope of this paper to present and evaluate these features. They are part of the stand-alone model LPJmL5 and are individually published in the referenced papers which we also cite in our paper. Our aim is to show that the coupling between LPJmL5 and the ocean - sea ice - atmosphere components works, gives reasonable results in climate and important vegetation variables, as well as the climatic stability, which opens possibilities for new research. For more information about atmosphere or ocean Earth system components, specific processes such as fire dynamics, the handling of crop growth and harvest in LPJmL5 we refer to the individual papers where these modules or features are described in greater detail. The current paper informs about the steps to facilitate a successful coupling of a DGVM to an Earth system model. The significance of our resulting model is raised in the introduction and described in more detail in the discussion (4.3).

The coupling of LPJmL5 to CM2Mc is motivated by the option to study climate feedbacks arising from changes in land processes such as dynamic vegetation, process-based fire, permafrost and an advanced land use scheme with explicit simulation of crop growth, incl. sowing and harvesting as well as managed grassland. The new coupled system allows us to investigate the detailed response of the biosphere to the combined response to climate and land-use change, while being computationally efficient. Other ESMs, such as CLM5 (Lombardozzi et al. 2020,https://doi.org/10.1029/2019JG005529), show that more and more ESMs are moving in that direction. Ongoing model intercomparison outline also that diversity in modelling approaches is helpful for charting possible future directions of our Earth system. Our evaluation results give us confidence in the robustness of our results. Detailed responses to your individual comments can be found inserted below.

1Individual comments

L89–94 It seemed that the limited variables are interconnected

Here, we are not quite sure what the reviewer's concerns are.

In this section the FMS coupler is described, which standardizes the interfaces between the model components and handles the fluxes between them. Hence through the coupler, which has some functionality of a LSM, the different variables are interconnected. We provide the most important variables for the land side (humidity, roughness, albedo and temperature) in the fast time step and the coupler calculates, e.g., the drag or the flux of moisture to the atmosphere. These variables are then provided to the land model and the atmosphere in the next time step. Fig. 1 aims to illustrate the FMS functioning. We hope this sufficiently addresses the reviewer's concern.

L89–94 I'm not familiar with ESMs. So, for me, it is hard to imagine how a coupler work between models. How do you reconcile the conservation of energy and mass as variables are exchanged between models with different resolutions?

We thank the reviewer for noticing here a lack of more detailed description. We added a few sentences to 2.1.1:
"*All model components are simulated on different spatial and temporal scales and the coupler is the interface directly connected to the different parts. It interpolates the different scales to a common grid and adapts the respective fluxes to the grid of the receiving model*

*component. Usually the variables are not directly exchanged between model components. For instance, the land model calculates the humidity of the canopy layer, and the atmosphere the humidity of the lowest atmospheric layer. The coupler calculates the moisture flux between both layers and provides them to the different models on their respective spatial and temporal scale, while the different humidity variables are not exchanged. By tracking these explicit fluxes of energy and water, the coupler ensures the conversation of these quantities.*"

2.1.2 & 2.1.3 Please don't use abbreviations in the section title. Like 2.1.1 or more insightful title is better.
2.2 Same above.
Thanks for pointing this out, indeed it helps to improve readability. We changed it in the revised document to Modular Ocean Model 5 (MOM5) and Atmospheric Model 2 (AM2).

L105 What are the input variables for this component. Please specify them as in L118.
This paper is about the coupling interface between the land model and the FMS coupler. As BLING is a component inside the ocean it is beyond the scope of this paper. Further information about BLING can be found in Galbraith et al. (2010). We improved the paper text to clarify this in 2.1.2:

*"Enclosed in the ocean component MOM5, the Biogeochemistry with Light, Nutrients and Gases (BLING) model is run. It was developed at Princeton/GFDL as an intermediate-complexity tool to approximate marine biogeochemical cycling of key elements and their iso-topes. For further details we refer to Galbraith et al., 2011."*

L110 dynamic core of ?
The dynamic core in the atmospheric model as presented in Lin et. al, 2004. In the revised manuscript we changed the sentence to:
*"It uses the finite volume dynamical core as in (Lin, 2004),..."*

L111 Sorry, what is a C and D grid. Perhaps, it is a very common term in the climate model study field.
While they are indeed very common terms, we deleted the remark about the C and D grids, since the atmosphere grid structure is mostly irrelevant for our coupling work. Interested readers are referred to the more detailed description in the referenced papers. We thank the reviewer for pointing to this possibility of clarifying the text. We rewrote this section as stated in the next remark.

L112 What do the tracers mean? What do the dynamics suggests?

Tracers are conservative or non-conservative properties (e.g. humidity), that are transported (advected or diffused) between grid cells.

Dynamics refer to the motion and thermodynamic state in the atmosphere. We agree with the reviewer, and rewrote this section to improve clarity. It reads now:

*"The atmospheric module in CM2Mc is GFDL's Atmospheric Model version 2.1 (AM2, Anderson et al. 2004). It uses the finite volume dynamical core as in Lin (2004), as implemented in CM2.1 (Delworth et al., 2006) with a latitudinal resolution of 3° and a longitudinal resolution of 3.75° and 24 vertical levels, the lowest being at 30m and the top at about 40 km above the surface. For the coupled setup, we use a general atmospheric time step of 1 hr at which variables are exchanged with the coupler. Dynamic motion and the thermodynamic state of the atmosphere are calculated on a 9 min time step, while the radiation scheme has a time step of 3 hrs. The coupled model includes an explicit representation of the diurnal cycle of solar radiation. For a more detailed description of the model and its configuration, see Galbraith et al. (2011) and Delworth et al. (2006)."*

L121 LPJmL5 -> LPJmL ver. 5? Please add the citation for this version.

The citation for LPJmL5 is in the first sentence of the presentation of LPJmL 2.2: Von Bloh et al. 2018.

All model versions that build on the LPJmL5 version published by von Bloh et al. 2018 have the nitrogen cycle implemented which can be deactivated in the model code to simplify a study. We realize that the formulation in lines 121-22 were perhaps misleading. The sentences now read:

*"All LPJmL (sub-)versions that build on the LPJmL5 version published by von Bloh et al. (2018), include the nitrogen and nutrient cycle. Because further adaptations would be necessary to include the nitrogen cycle in the coupled model, we concluded that it is beyond the scope of this study and deactivated it in this study."*

L126 Please summarize all plant functional types in the manuscript or the supplemental material. Although the simulation results are presented in Fig 8, the readers can only reach the abbreviation about PFT.

We thank the reviewer for this useful suggestion and added a list of all plant functional types to the Appendix B.

L131 Which dataset was used for the prescribed land-use input?

This information is given in Section 2.4, Model setup and forcing.

*"...and land-use informationare from Fader et al. (2010)"*

2.3.2 Section 2.3.2 should be incorporated into Section 2.2 (LPJmL part).

We disagree with this suggestion, because Section 2.3.2 is the most important and core section of the methods used for the coupling work. It explains the changes to LPJmL in order to prepare the model for the coupling. Section 2.2 just gives an overview of the current stand-alone model LPJmL to provide the reader with the basic information on the main functionalities of the DGVM. To couple this model to CM2Mc we had to implement several

new features, including a new canopy module with a calculation of the energy balance and humidity-driven evapotranspiration, which are not part of the standard LPJmL model. We added a statement in the beginning of 2.3. to clarify this:

*"While Section 2.2 described the standard LPJmL5 model as previously published we introduce in Section 2.3 our adaptations to LPJmL5 in order to be coupled with the FMS coupling framework."*

L174–179 It is better to describe this sentence in the introduction.
We agree that part of this information is important for the introduction. We wrote a similar, more general statement in the introduction in L 51-55:

*"With increasing process-detail and the number of processes captured in the biosphere components of ESMs rising, new challenges in correctly representing potential feedback mechanisms might arise. This includes error propagation resulting from changes in climate that could be amplified by, e.g., increased tree mortality, which then changes land-surface characteristics over time (Quillet et al., 2010). Hence, a bidirectional and stable coupling of a DGVM with a full water, energy and carbon cycle remains a challenge (Forrest et al., Pokhrel et al. 2016)"*

We think, however, that coming back to this point in a more detailed manner is important to explain and contextualize the changes to LPJmL also in the methods. Part of this paragraph in L174-179 is too detailed and presumes more context than the introduction can provide.

L321 Please add the citation for the "historic land-use data from 1700".
Although it was also before stated in 2.4 "Model setup and forcing", we follow the recommendation and add the citation in the revised manuscript here as well. .
*"...prescribed as described in Fader et al. (2010)"*

L354 Please provide here which variables y i are evaluated by this metric.
Section 2.6 is structured in a way that in the first part the different variables which are evaluated are presented and in the last part the used metric is presented. Hence $y_i$ signifies all the variables introduced before: temperature, precipitation, vegetation. For clarification we added a sentence in the end of the chapter:

*"We use this metric for the evaluation of the performance of temperature, precipitation and above ground biomass."*

L367: 3.1.1 How about the stability in other variables in other model components such as PFT distribution, C stock in terrestrial and Ocean ecosystems?

This is an interesting question. Since this paper is mostly about the description of the coupling approach and a basic evaluation of the biosphere, we regard the global mean temperature as a sufficient indicator for a stable equilibrium between climate and land biosphere. With a stable climate, as shown in the paper, it is assumed that other internal model variables are also getting stable.

The reviewer raises an interesting point on the stability of the terrestrial carbon stock. Since we did large modifications on the biophysical part of LPJmL, and the DGVM is now running in a coupled mode, we agree it is a good idea to also provide a curve of total carbon in the biosphere for 1000 modeling years, which can be found now in the supplement. Large changes in the carbon pool integrate all changes in the PFT distribution and composition as well as short term reactions such as fire disturbance.

The evaluation of the stability of C stocks in the ocean is out of scope for this paper, since we did not do any changes in the ocean model and have a strong focus on the terrestrial biosphere. The approach of our evaluation is explained in the last paragraph of the Introduction (L72-76). To discuss the relationship between a stable climate and generally stable stocks in the different model components we added to 4.1:

*"By achieving a stable climate in terms of surface temperature and precipitation, other variables in the model as for instance carbon stocks of the biosphere (see Fig. S8 in the Supplement) and ocean carbon stocks are also assumed to stabilize (even though possibly on a different time scale)."*

[Figure]

Fig. 1: Total carbon in the biosphere (vegetation, litter, soil) for the piControl run.

L410 Are there any improvement in the CM2Mc-LPJmL output that should be mentioned, compared to CM2Mc-LaD?
The explicit comparison between CM2Mc-LPJmL and CM2Mc-LaD is available in the Supplement S3. Generally the biases in CM2Mc-LPJmL are slightly larger.
LPJmL, however, is an advanced DGVM which includes vegetation dynamics, permafrost dynamics, fire disturbances and a comprehensive managed land module, which are feedbacks, which might increase instabilities in the model results. Using a dynamic vegetation model opens up research possibilities about interaction/feedback/etc , which are not possible with either the simple LaD nor stand-alone LPJmL. We added to 3.1.3 and 3.1.4 a reference to the figures in Supplement Section S3.

2L414 & Figure 6 I don't know the manner of climate science, but why do you compare annual precipitation in mm/day instead of mm/year? It is hard to intuitively understand the size of the bias.

The basis of the data in Figure 6 are daily values, which are presented in a mean from 1994-2003. Hence, it makes sense to show the results in units of mm/day, as done in other studies (e.g. Anderson et al. 2004).

L450– There are no information about PFTs in detail. I can see only abbreviation in the legend of Figure 8. So, please add this information in material and method or in appropriate place.

We thank the reviewer for this useful suggestion and added a list of all plant functional types to the Appendix B.

L485 If the results of the comparison with CMIP5 are important, it would be better to include them as a figure in the draft instead of sending them to the supplement.

In our opinion, these comparisons are useful but not very important, because the direct comparison between our model and CMIP5 models is difficult. These models employ a much more detailed atmosphere and a much finer grid-size, resulting in better constrained climate and vegetation. While a direct comparison is difficult, our results are, however, in the range of CMIP5 which is an important plausibility check for our new coupled system. We therefore believe that putting these comparisons to the supplement is sufficient while maintaining the clarity of the main text.

L587–590 Some of this text could be taken to the intro and explained as to why we are creating a new ESM using the coupler.

This is a good suggestion, but to our opinion, similar information as to why we are coupling LPJmL to CM2Mc is stated in the introduction:

*"Benefits of coupling LPJmL5 include the use of the process-based fire model SPITFIRE (Thonicke et al., 2010; Drüke et al., 2019), its advanced land use and land management scheme, the representation of permafrost and a state-of-the-art water cycling (Schaphoff et al., 2018a). By using FMS as the coupling infrastructure we remain flexible in terms of other ESM components. The coarse CM2Mc model grid enables us to have a relatively fast and computationally low-cost Earth system model, which allows conducting many model realisations under different land use and trace gas settings. While CM2Mc uses the relatively old, but fast atmospheric model AM2 (Anderson et al., 2004) in a coarse resolution setup and the ocean model MOM5 (Galbraith et al., 2011), it will be possible to employ the latest GFDL model developments in our coupled system in the future."*

Figure 1 Figure 1 is not referred in the text. Are there any exchange between MOM and LPJmL? How about atmospheric pressure between AM2 and LPJmL.

We thank the Reviewer for noting this. We now refer to this Figure in section 2.3. In the current implementation there is no direct exchange between MOM and LPJmL, but of course

several indirect links exist via, e.g., exchange of heat, changes in the sea surface temperature, which influence climate and thus  land surface and vegetation.

It is true that we are using surface pressure, provided by the coupler. There are however no direct or large feedbacks between vegetation and surface pressure, hence pressure is much less important in this context as, e.g., temperature and humidity. For clarification we added the most important variables to the caption of Figure 1:

*"Schematic overview of CM2Mc-LPJmL and the most important variables exchanged between LPJmL5, FMS and AM2."*

---

## Referee Report (RR1)

Dear authors,

thank you for your thorough revision of the manuscript. The model is much clearer now. However, I still have two minor comments.

1) The Penmann Monteith Equation, as it appears in the manuscript, is:

$$\lambda ET_0 = \frac{\frac{dq_{sat}}{dT}(R_n - G) \cdot +86400 \cdot \frac{\rho_a C_p(e_s^0 - e_a)}{\tau_{av}}}{\frac{dq_{sat}}{dT} + \gamma \left(1 + \frac{\tau_s}{\tau_{av}}\right)}$$
(1)

This equation still needs revision. Plugging in the values and units that you specify in your manuscript right after the equation, the term

$$\frac{dq_{sat}}{dT}(R_n - G) \tag{2}$$

has units of kPa°C-1Jm-2s-1. Since the factor 86400 is in s/day, and  $C_p$  is in MJkg-1°C-1, the second term in the numerator,

$$86400 \cdot \frac{\rho_a C_p (e_s^0 - e_a)}{\tau_{av}}$$
 (3)

has units of  $kPa^{\circ}C^{-1}MJm^{-2}day^{-1}$ .

The units of these two terms must match. The energy is expressed in J in one term and in MJ in the other. Time is expressed in days in one term and in seconds in the other. In your response to my first comment you state that the daily version of the equation is used, but then divided by the number of time steps in one day. I guess what you mean is that you convert the subdaily values of  $R_n$  and G from W/m2 to J/day so that the 86400 factor makes sense, and then divide the resulting  $\lambda ET_0$  by 24 and convert back to seconds? If there's an 86400 factor, the result of the calculation would be in MJm-sday-1, but this cannot be the case if  $ET_0$  on the left hand side of Eq. (1) is in mms-1, as is stated in the manuscript. Please, check the units and conversion factors of this equation.

Also, note the typo on the numerator, there's a multiplication symbol  $(\cdot)$  just before the +86400.

2) It is stated that "While the new potential evapotranspiration is calculated in the subdaily time step, the non-water-stressed canopy conductance is calculated in a daily time step, due to the daily calculation of the photosynthesis in LPJmL5". If you are using daily values of the non-water-stressed canopy conductance I guess you use the cumulative net PAR, temperature, etc... at the end of the day to calculate the canopy conductance for the next day? I.e., the  $g_p$  you use one day is calculated with the values (PAR, temperature, ...) from the previous day? This should be clear in the text. Also, what temperature are you feeding now to the photosynthesis routine? A diurnal average of the canopy temperature, or the air temperature?

Best regards,